# Generation of adult hippocampal neural stem cells occurs in the early postnatal dentate gyrus and depends on cyclin D2

Oier Pastor-Alonso [1,7,8], Anum Syeda Zahra [2,8], Bente Kaske[2], Fernando García-Moreno[1,3,4], Felix Tetzlaff[2], Enno Bockelmann [2], Vanessa Grunwald[2], Soraya Martín-Suárez[1], Kristoffer Riecken[5], Otto Wilhelm Witte[2,6], Juan Manuel Encinas [1,3,4 ✉] & Anja Urbach [2,6 ✉]

## Abstract

**Lifelong hippocampal neurogenesis is maintained by a pool of multipotent adult neural stem cells (aNSCs) residing in the subgranular zone of the dentate gyrus (DG). The mechanisms guiding transition of NSCs from the developmental to the adult state remain unclear. We show here, by using nestin-based reporter mice deficient for cyclin D2, that the aNSC pool is established through cyclin D2-dependent proliferation during the first two weeks of life. The absence of cyclin D2 does not affect normal development of the dentate gyrus until birth but prevents postnatal formation of radial glia-like aNSCs. Furthermore, retroviral fate mapping reveals that aNSCs are born on-site from precursors located in the dentate gyrus shortly after birth. Taken together, our data identify the critical time window and the spatial location of the precursor divisions that generate the persistent population of aNSCs and demonstrate the central role of cyclin D2 in this process.**

**Keywords** Adult Neurogenesis; Cyclin D2; Dentate Gyrus; Development; Neural Stem Cells
**Subject Categories** Neuroscience; Stem Cells & Regenerative Medicine

## Introduction

The dentate gyrus (DG) is one of the few brain areas where neurogenesis persists into adulthood. The newly born neurons integrate and contribute to hippocampal functions including learning, memory and mood regulation, and alterations in their production have been associated to mental and neurological diseases (Toda et al, 2019). The source of this process lies in a pool of quiescent adult neural stem cells (aNSCs), which,

upon activation, undergo mostly asymmetric division to self-renew and give rise to new neurons (Encinas et al, 2011; Kempermann, 2015). While symmetric NSC division is also possible (Bonaguidi et al, 2011), the division-coupled depletion of aNSCs outweighs their capacity for self-renewal, leading to a decline in neurogenesis with age (Encinas et al, 2011; Ibrayeva et al, 2021; Pilz et al, 2018). Consequently, the neurogenic capacity of the adult DG is largely determined by the initial size of the aNSC pool established during development.

The actual origin of aNSCs and the mechanisms controlling their establishment in the DG remain poorly understood. Adult NSCs reside in a special abventricular niche, the subgranular zone (SGZ), which is formed by postnatal day (P) 14 in mice (Nicola et al, 2015; Seki et al, 2014). Morphogenesis of the DG starts at late gestation and continues into postnatal periods (Altman and Bayer, 1990b; Nelson et al, 2020; Seki et al, 2014). A key feature of DG development is the formation of a dentate migratory stream (DMS) by developmental NSCs (dNSCs) and their progeny, which detach from the dentate neuroepithelium (DNe) to form multiple transient niches and ultimately the DG (Altman and Bayer, 1990a; Li et al, 2009). Although dNSCs enter the nascent DG at embryonic stages, forming new hilar and subpial precursor niches inside the DG, the DMS remains active during the early postnatal period (Altman and Bayer, 1990a; Hodge et al, 2012).

Different studies suggest that aNSC precursors migrate along the DMS during embryonic development before colonizing the SGZ, where they persist life-long (Berg et al, 2019; Hodge et al, 2013; Li et al, 2009). Others suggest that Sonic hedgehog-responsive precursors originating from the ventral hippocampus at E17.5 serve as a source of aNSCs (Li et al, 2013; Noguchi et al, 2019). Genetic fate mapping studies using HopxCreER[T2] mice showed that aNSCs and dNSCs share a common lineage and continuity in fate specification, suggesting that adult neurogenesis is a continuation of development (Berg et al, 2019). Although transcriptome analysis of the developmental and adult neurogenic cascade supported this perspective, the molecular profiles of aNSCs and dNSCs were

[1]Laboratory of Neural Stem Cells and Neurogenesis, Achucarro Basque Center for Neuroscience, Scientific Park, 48940 Leioa, Bizkaia, Spain. [2]Department of Neurology, Jena University Hospital, 07747 Jena, Germany. [3]IKERBASQUE, The Basque Foundation for Science, Plaza Euskadi 5, 48009 Bilbo, Bizkaia, Spain. [4]Department of Neurosciences, University of the Basque Country (UPV/EHU), Scientific Park, 48940 Leioa, Bizkaia, Spain. [5]Research Department Cell and Gene Therapy, Department of Stem Cell Transplantation, University Medical Center Hamburg-Eppendorf, 20246 Hamburg, Germany. [6]Jena Centre for Healthy Aging, Jena University Hospital, 07747 Jena, Germany. [7]Present address: Department of Neurology, University of California San Francisco, San Francisco, CA 94143, USA. [8]These authors contributed equally: Oier Pastor-Alonso, Anum Syeda Zahra. ✉E-mail: jm.encinas@ikerbasque.org; anja.urbach@med.uni-jena.de

significantly different (Berg et al, 2019; Hochgerner et al, 2018; Matsue et al, 2018; Valcarcel-Martin et al, 2020). Regardless of their embryonic origin, early postnatal cell division has been suggested to play a key role for the formation of aNSCs (Ortega-Martinez and Trejo, 2015; Youssef et al, 2018), supporting the idea that rather than being mere remnants of the developmental population, aNSCs may be a distinct population that emerges from dNSCs during a critical postnatal stage.

Previous work suggests that adult hippocampal neurogenesis requires cyclin D2, one of three D-cyclins essential for the progression of cells through the G1 restriction point in response to mitogens (Ansorg et al, 2012; Kowalczyk et al, 2004). We showed that this requirement builds up progressively during the early postnatal period, culminating in a proliferative arrest in cyclin D2-deficient mice between P14 and P28 (Ansorg et al, 2012). The fact that this period coincides with the formation of the aNSC pool and that the proliferative deficit of these mice cannot be overcome by exposure to neurogenic stimuli (Jedynak et al, 2012; Kowalczyk et al, 2004) led us to hypothesize that cyclin D2 might be relevant in the establishment of the aNSC pool. Here, we used cyclin D2 knockout (D2KO) mice and targeted in vivo injection of retroviral vectors to describe when, how and where dNSC precursors divide and give rise to long-lived aNSCs. We found that cyclin D2-dependent proliferation is crucial for the postnatal formation of aNSCs and that the final mitosis generating aNSCs takes place on-site, in the DG during the first and a half week of life.

# Results

## Inactivation of cyclin D2 depletes the population of self-renewing aNSCs

Previous studies suggest that cyclin D2 is critical for maintaining adult neurogenesis (Ansorg et al, 2012; Kowalczyk et al, 2004), but its function in aNSCs remains largely unclear. To specifically investigate the role of cyclin D2 in aNSCs, we used D2KO and WT littermates expressing GFP under control of the nestin promoter. In contrast to the densely populated WT SGZ, the SGZ of D2KO mice was virtually devoid of GFP$^+$ cells with a radial glia-like phenotype (Fig. 1A). Most of the remaining GFP$^+$ cells displayed a complex morphology. To quantify these differences in detail, we performed an unbiased 3D-Sholl analysis on randomly selected GFP$^+$GFAP$^+$ cells. While the Sholl profiles of WT aNSCs reflected the prototypical morphotype with a single primary process and distal ramifications (Fig. 1B), aNSCs from D2KO mice displayed more proximal intersections and a smaller maximal radius (Fig. 1B,C), confirming their higher complexity.

To determine whether the numerical and morphological changes in mutant aNSCs reflect an imbalance in the composition of the aNSC pool, we classified GFP$^+$GFAP$^+$ aNSCs into three previously described subtypes (Fig. 1D; Gebara et al, 2016; Martin-Suarez et al, 2019). Confirming these earlier reports, most aNSCs of WT mice were radial glia-like α-cells (Fig. 1E,F). The D2KO led to a disproportionate loss of α-cells (Fig. 1E), that were almost absent in absolute terms (Fig. 1F). Instead, most mutant aNSCs were Ω-cells (Fig. 1E,F), known to represent a deeply quiescent aNSC subtype (Martin-Suarez et al, 2019). Thus, we co-stained against Ki67 and confirmed that α-cells represent the

majority of dividing aNSCs regardless of genotype, while Ω-cells rarely divide (Fig. 1G). Accordingly, the SGZ of D2KO mice was virtually devoid of actively dividing cells (Ansorg et al, 2012; Fig. 1H).

To corroborate these in vivo data, we performed neurosphere assays with serial passaging which help to address mitotic potential in vitro. Although neurospheres can be derived from aNSCs and their transit-amplifying progeny, only true NSCs are capable to self-renew over extended periods of time (Reynolds and Rietze, 2005; Walker and Kempermann, 2014). We found that cultures derived from young adult D2KO mice formed significantly less and smaller primary spheres than those of WT mice (Fig. 1I,J), consistent with a numerical and functional deficit in aNSCs. When subcultured, mutant cells failed to expand and exhausted after a few passages (Fig. 1K), confirming the lack of self-renewing aNSCs in the D2KO. Together, our in vivo and in vitro data demonstrate that the absence of cyclin D2 leads to selective loss of the radial glia-like, self-renewing population of aNSCs.

To further evaluate the role of cyclin D2 in the adult neurogenic lineage, we examined the expression of cyclin D2 in the adult WT SGZ (Figure EV1A–G). Although all precursor types from aNSCs to neuroblasts were represented in the cyclin D2$^+$ cell population, it mainly comprised transit-amplifying stages (Figure EV1G). Moreover, cyclin D2 was found to be the predominant D-cyclin of aNSCs (Fig. 1L–N) compared with cyclin D1, the other D-cyclin expressed in the adult SGZ (Figure EV1B,C,E). Co-staining with MCM2 revealed that the majority of cyclin D2$^+$ cells were actively dividing, while the cyclin D1$^+$ population was mostly quiescent (Figure EV1H–J), altogether suggesting that cyclin D2 is required for lineage amplification during early stages of adult hippocampal neurogenesis.

## Cyclin D2 is expressed in dNSCs

Although the results obtained in adult WT mice indicate that cyclin D2 is important for aNSC activation, the severity of the D2KO phenotype actually suggests a requirement for the formation of the aNSC pool. Cell division during early postnatal development has been suggested to play a role in the establishment of the aNSC population (Ortega-Martinez and Trejo, 2015; Youssef et al, 2018). Hence, we examined the expression of D-cyclins in radial glia-like dNSCs and aNSCs in the SGZ or perinatally, when NSCs occupy the SGZ and the GCL, throughout the entire SGZ/GCL. We analyzed five postnatal stages (P0, P7, P10, P14 and P28), covering the period from when the DG is highly immature and NSCs are not considered adult (dNSCs; P0) to the age when NSCs with adult properties, such as quiescence, distinctive morphology and expression of unique biomarkers such as lysophosphatidic receptor 1 (aNSCs; Valcarcel-Martin et al, 2020), are restricted to the SGZ (P14 onwards; Berg et al, 2019; Nicola et al, 2015; Figs. 2 and EV2). We found that cyclin D2 is expressed by NSCs at all developmental stages, especially during the first 10 days of life when most of them were cyclin D2$^+$ (Fig. 2B,C). After a peak of expression at P7, the proportion of cyclin D2$^+$ NSCs continuously declined, reaching the low adult level at P28 (Fig. 2C). Further quantification of cyclin D2$^+$ and D2$^-$ NSC numbers revealed that cyclin D2$^+$ NSCs appear only transiently, increasing significantly from birth to P10, before declining and returning to nearly P0 levels at P28 (Fig. 2D). Next, we examined cyclin D1, which is also expressed by aNSCs

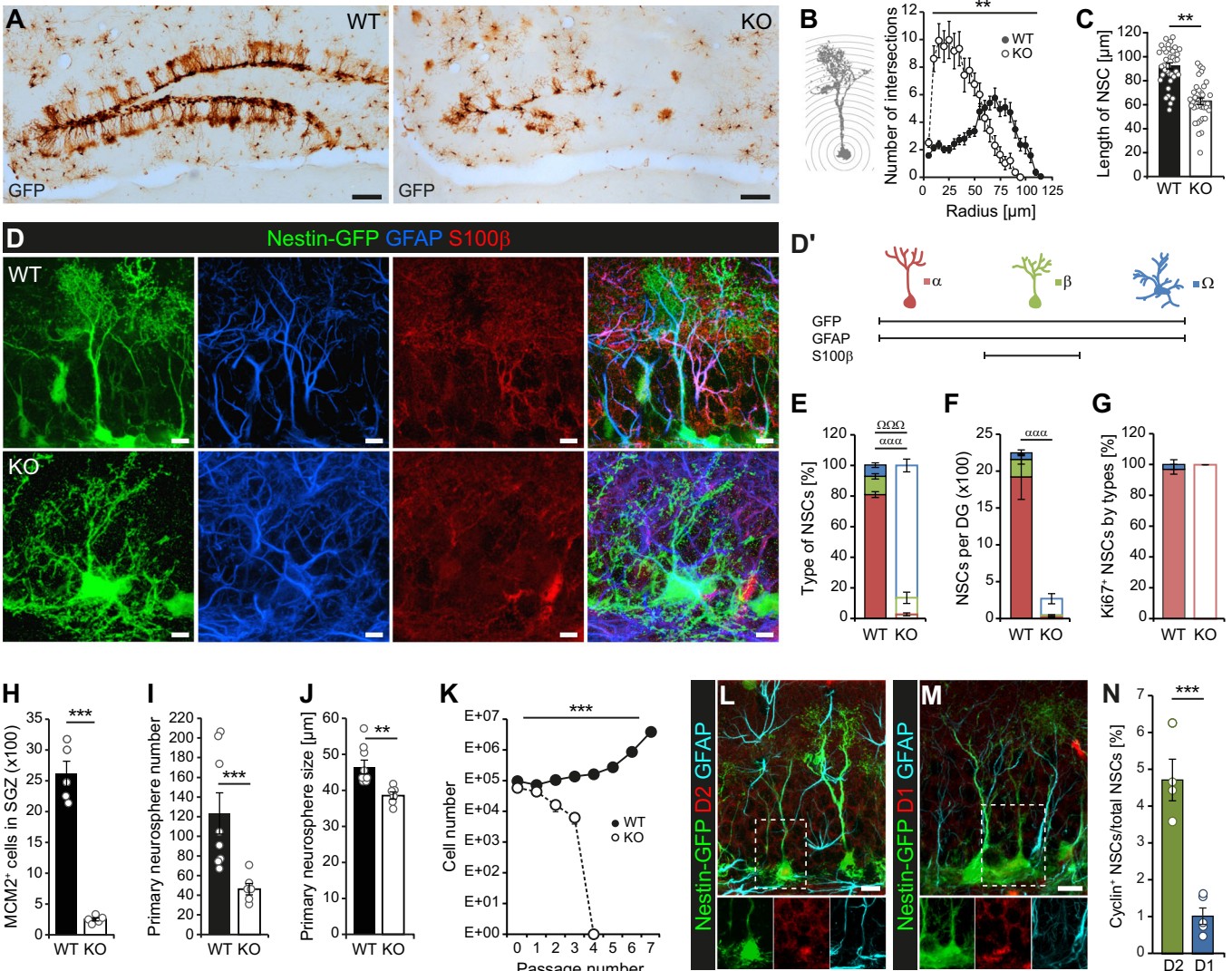

**Figure 1. Lack of cyclin D2 results in a loss of NSCs in the adult DG.**

(A) ABC peroxidase staining of GFP in adult WT and D2KO mice illustrating the lack of nestin expressing aNSCs in D2KO mice. Scale bar 100 μm. (B, C) 3D-Sholl analysis of randomly selected GFP[+] aNSCs ($n = 4$ mice/group, 8-10 cells/mouse) demonstrating increased complexity (B) and decreased length (longest axis with hilus-to-molecular-layer orientation; C) of mutant aNSCs. (D) Confocal images of GFP, GFAP and S100β immunostained aNSCs illustrating the phenotypical differences between prototypical aNSCs in WT and aNSCs of D2KO mice. Scale bar 10 μm. (D´) Schema summarizing the criteria applied to distinguish α, β and Ω cells. (E, F) Classification of aNSC types based on the expression of S100β and morphological criteria showing the percentage (E) and total (F) changes of aNSC pool composition in D2KO mice ($n = 4$ mice/group). (G) Proportion of aNSC types within the dividing (Ki67[+]) aNSCs population ($n = 4$ mice/group). (H) Quantification of actively dividing (MCM2[+]) cells in the SGZ of D2KO and WT mice ($n = 5$ mice/group). (I, J) Neurosphere-forming capacity of the DG during primary culture (cultures from $n = 8$ WT mice and $n = 6$ KO mice). (K) Long-term self-renewal capacity of WT and D2KO cultures (cultures from $n = 3$ WT mice and $n = 4$ KO mice). (L–N) Confocal images (L, M) and quantification (N) of cyclin D2[+] and D1[+] aNSCs ($n = 4$ for cyclin D2 and $n = 5$ for cyclin D1). Scale bar 10 μm. Data information: All values represent mean ± SEM. Statistics: 2-way RM-ANOVA (B, E–G, K), Student's $t$ test (C, J, N), Welch's $t$ test (H), Mann–Whitney $U$-test (I); *$P < 0.05$, **$P < 0.01$, ***$P < 0.001$; $^{ααα}P < 0.001$ for α cells; $^{ΩΩΩ}P < 0.001$ for Ω cells. Source data are available online for this figure.

(Fig. 1M,N) but appears largely dispensable for adult neurogenesis (Kowalczyk et al, 2004). Except for P0, when more NSCs expressed cyclin D1 than cyclin D2 (Fig. 2C), the proportions and absolute numbers of cyclin D1[+] NSCs followed a similar time course as those expressing cyclin D2 (Figs. 2C,E and EV2). The high expression of both D-cyclins in dNSCs indicates a substantial overlap of expression. Therefore, we calculated a minimum co-expression index for each age and determined the zero of the curve

fit. This approach revealed that cyclins D2 and D1 must be co-expressed in a fraction of dNSCs until at least P13 (Fig. 2F).

In parallel to the D-cyclin[+] NSCs, but with a slight delay, the populations of cyclin D[-] NSCs increased from P7 and continued expanding until P14, after P28 they began to decline (Fig. 2D,E). These data demonstrate that both D-cyclins are highly expressed by dNSCs during the time of peak proliferation in the developing SGZ (Altman and Bayer, 1990a; Bond et al, 2020) and are downregulated

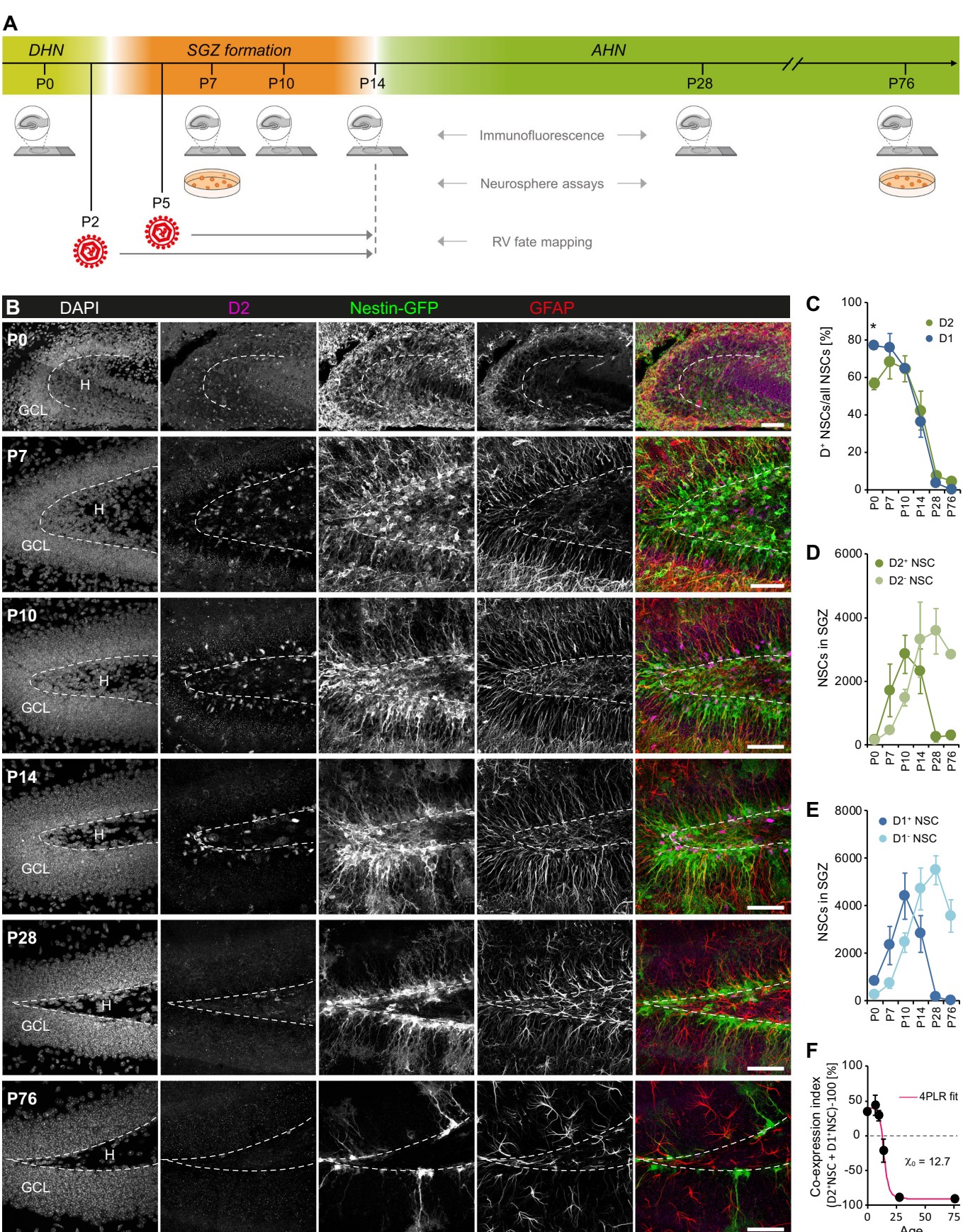

**Figure 2. Cyclin D2 as well as cyclin D1 are expressed in NSCs of the peri- and postnatal DG.**

(A) Scheme of the study design for investigating the SGZ niche in WT and D2KO mice. (B) Confocal images illustrating the expression of cyclin D2 in the hilar and SGZ niche of the developing DG. Images represent maximum intensity projections of 12.07 µm high Z-stacks. Scale bars 50 µm. GCL, granule cell layer; H, hilus. (C) Proportions of NSCs expressing cyclin D2 or cyclin D1 (D2: $n = 4$ mice/group; D1: $n = 4$ mice/group except P14 with $n = 5$). See also Figure EV2. (D) Total number of cyclin D2$^+$ and D2$^-$ NSCs ($n = 4$ mice/group). (E) Total number of cyclin D1$^+$ and D1$^-$ NSCs ($n = 4$ mice/group except P10 KO and P14 WT with $n = 5$). See also Figure EV2. (F) Co-expression index of cyclin D2 and D1 in NSCs to estimate how long their expression must overlap in NSCs. Curve fit represents a Four Parameter Logistic (4PL) Regression. $N = 4$ mice/group except P0 with $n = 3$. Data information: All values represent mean ± SEM. Statistics: 2-way ANOVA, *$P < 0.05$. Source data are available online for this figure.

in parallel with the transition of NSCs to a quiescent adult state (Berg et al, 2019; Fig. 3B,C).

## Cyclin D2 deficiency impairs the formation of the aNSC pool

Next, we examined the development of the aNSC pool in D2KO mice. We previously showed that overall proliferation is reduced in the SGZ of D2KO mice as early as P7 (Ansorg et al, 2012). To evaluate whether the D2KO alters the cycling activity of dNSCs, we co-stained them with Ki67 (Fig. 3A) and analyzed the SGZ, since this is the anatomical endpoint of NSCs during DG development and the area where aNSCs are located. Interestingly, the proportion of dividing NSCs in the D2KO was unaltered (Fig. 3B). However, in total numbers, significantly fewer Ki67$^+$ NSCs were found in the DG of D2KO mice from P10 onwards (Fig. 3C), suggesting that the number of NSCs produced, rather than their propensity to divide, was decreased due to the absence of cyclin D2. Accordingly, quantification of total NSCs in the SGZ revealed a progressive deficit in NSC pool amplification from P7 onwards (Fig. 3D). Consistent with other studies (Glickstein et al, 2007a; Kowalczyk et al, 2004), the overall architecture of the mutant DG at perinatal age was comparable to that of WT mice. Moreover, at P0, the number of dNSCs was similar in both groups, indicating that embryonic development of the DG and its germinative niches is not significantly affected by the deletion of cyclin D2. From P0 to P7, the number of NSCs increased in both WT and D2KO mice, although tending to be higher in WT mice. Between P7-P10, when NSCs transition from developmental to adult, WT mice displayed the greatest NSC population growth (Fig. 3D,E) which failed to occur in D2KO mice (Fig. 3D). After P10, NSCs of KO mice gradually disappeared, increasing the difference to WT further (Fig. 3D). From P14 onwards, a decline in aNSC numbers was noticeable also in WT mice (Fig. 3A,D), suggesting that the balance between aNSC generation and depletion had already tipped in favor of exhaustion. Further investigation of the mutant NSC pool revealed that the transient cyclin D1$^+$ dNSC population, albeit in smaller numbers, is formed also in D2KO mice (Figure EV2B,D), whereas the expansion of the cyclin D1$^-$ population was completely prevented (Figure EV2C). This suggests that cyclin D1 contributes to dNSC proliferation but cannot compensate for the lack of cyclin D2 during aNSC amplification. The impaired formation of the aNSC pool observed in mutant mice was accompanied by a reduced growth of the SGZ, resulting in a significantly smaller SGZ from P10 onwards (Fig. 3F). These data suggest that cyclin D2 becomes increasingly required for NSC divisions and DG neurogenesis during the first two weeks of life, whereas it is largely dispensable before birth. This was confirmed in neurosphere assays with NSCs isolated from the P7 DG (Fig. 3G–I). Even though cultures of D2KO mice produced

fewer primary neurospheres (Fig. 3G), their size was similar (Fig. 3H) and they were able to grow exponentially (Fig. 3I), demonstrating the presence of self-renewing NSCs in the early postnatal DG of mutant mice.

To examine whether the lack of aNSCs in the mutant DG was indeed caused by a deficit in their generation instead of premature differentiation or death of dNSCs, we quantified the densities of astrocytes, neuroblasts and apoptotic cells in the different layers of the developing DG (Figure EV3). GFAP$^+$ stellate astrocytes of different developmental stages (immature: GFP$^+$, mature: GFP$^-$) were found to a similar extent in both groups, except for the postnatal germinative niches in the subpial zone and the hilus of D2KO mice, which contained fewer astrocytes in comparison to WT mice (Figure EV3A). The density of apoptotic cells in the mutant SGZ was actually smaller from P7 onwards, consistent with fewer NSCs and reduced neurogenesis, but similar to WT in the hilus (Figure EV3B). As well, the density of DCX$^+$ neuroblasts in the SGZ (Figure EV3C) and the volume of the GCL were smaller in KO mice (Figure EV3D). The finding that neither neurogenesis, astrogliogenesis nor cell death are increased during DG development of mutant mice supports the assertion that the D2KO directly impairs the generation of aNSCs.

Taken together, these data demonstrate the requirement of cyclin D2 for the late-stage cell divisions that lead to the formation of aNSCs in the early postnatal DG. Even though deficiency of cyclin D2 does not affect regular ontogenesis of the DG during the fetal period, it prevents the generation of aNSCs, resulting in premature loss of neurogenic capacity in the juvenile DG.

## Spatially restricted in vivo retroviral labeling reveals the postnatal origin of aNSCs

Next, we determined where exactly aNSC precursors divide and give rise to aNSCs. Through stereotactic injection of γ-retroviral vectors that carry the gene for the red fluorescent protein mCherry (SFFV-RV-mCherry) and exclusively transduce dividing cells into nestin-GFP pups, we specifically compared the contribution of precursors that divide while migrating along the DMS with those already located inside the postnatal DG (Altman and Bayer, 1990a; Berg et al, 2019; Hodge et al, 2013; Sugiyama et al, 2013). After validating the spatial precision of the DG and DMS injections (Figs. 4C and EV4A,B), we performed injections at P2 or P5, representing ages before and close to the peak generation of aNSCs (Berg et al, 2019; Ortega-Martinez and Trejo, 2015), and sacrificed the mice at P14, when the aNSCs pool is fully established. In all cases, mCherry$^+$ neurons with prototypical dendritic arborization extending into the molecular layer (ML) and axons extending into the hilus were found (Figs. 4A,B and EV4C,D). Strikingly, while mCherry$^+$ aNSCs were consistently found when RV-SFFV-mCherry was injected into the DG, only very few were detected

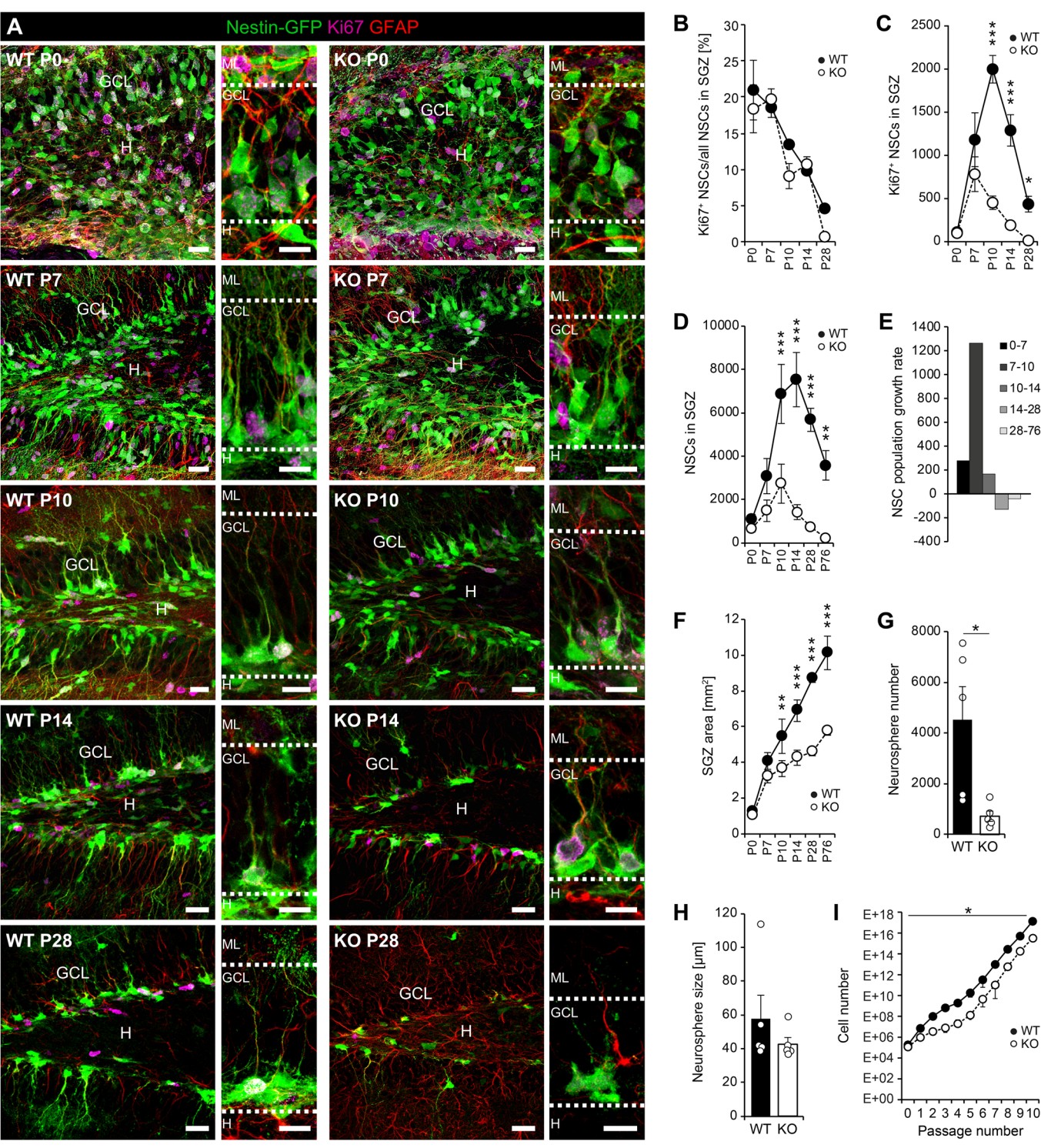

when injecting into the DMS at P2, and none when injecting at P5 (Fig. 4D,E). This indicates that the vast majority of aNSCs are generated from precursors located in the DG early after birth, while precursors dividing in the postnatal DMS hardly contribute to the formation of aNSCs and are almost exclusively neurogenic. Summarizing these results, we conclude that the persistent aNSC population is generated on-site in the DG via division of dNSCs that colonized the DG before P2.

## Discussion

Our study reveals the indispensable role of cyclin D2 in the generation of the hippocampal aNSC pool. The cyclin D2-driven formation of aNSCs during a discrete time window (P0–P14) and from specific precursors that enter the DG perinatally and differ in their potential to form aNSCs from those that continue to enter the DG via the DMS, suggests that albeit being a continuation of the

**Figure 3.  Absence of cyclin D2 impairs the generation of aNSCs.**

(A) Confocal images of nestin-GFP, Ki67 and GFAP expression in the developing DG of WT and D2KO mice. Images represent maximum intensity projections of 9–12 μm high Z-stacks. Scale bars 25 μm in lower magnification images, 10 μm in higher magnification images. (B, C) Percentage (B) and absolute number (C) of proliferating (Ki67$^+$) NSCs ($n = 4$ mice/group except $n = 3$ in P7 WT). (D) NSC quantification showing that the D2KO prevents the expansion of the aNSC population during the first 2 weeks of life ($n = 4$ mice/group except $n = 5$ in P10 KO and P14 WT). (E) Average NSC pool growth rate per day in WT, demonstrating the strongest expansion between P7 and P10. (F) Quantification of the SGZ area ($n = 4$ mice/group except $n = 6$ at P0 and $n = 5$ at P14). (G, H) Primary neurosphere-forming capacity of postnatal DG precursor cells isolated at P7 (cultures from $n = 5$ mice/group). (I) Long-term self-renewal capacity of P7 cultures (cultures from $n = 3$ mice/group). Data information: All values represent mean ± SEM. Statistics: 2-way ANOVA (B–D, F), Welch's $t$ test (G), Mann–Whitney $U$-test (H), 2-way RM-ANOVA (I); *$P < 0.05$, **$P < 0.01$, ***$P < 0.001$. The horizontal bar in (I) represents a main effect of genotype. Source data are available online for this figure.

developmental process (Berg et al, 2019), aNSCs and adult neurogenesis are distinct from their developmental counterparts.

Although D-cyclins are interchangeable in many cellular contexts (Sherr and Roberts, 2004), several lines of evidence suggest that they possess unique roles during nervous system development (Glickstein et al, 2007a; Glickstein et al, 2007b; Lukaszewicz and Anderson, 2011; Wianny et al, 1998). In the DG, deletion of cyclin D2 but not of cyclin D1 leads to a severe deficit in neurogenesis that builds up progressively during the early postnatal period (Ansorg et al, 2012; Kowalczyk et al, 2004). Because this period coincides with the appearance of aNSCs in the SGZ (Nicola et al, 2015; Ortega-Martinez and Trejo, 2015), we hypothesized that the impaired adult neurogenesis of D2KO mice was due to the failed generation of the population of aNSCs during DG development. Our results show that deletion of cyclin D2 indeed prevents the formation of the radial glia-like aNSC pool early on and that the few NSCs with adult features that are still born deplete much earlier than usual. This effect could not be compensated by cyclin D1, which, however, is sufficient to ensure prenatal morphogenesis of the DG and the proliferation of dNSCs in the perinatal DG. The cause for the differential requirement for D-cyclins in dNSCs and aNSCs has not been established, but may involve the preferential regulation of cyclins D1 and D2 by spatiotemporally changing niche signals and differences in their non-canonical effects on downstream molecular pathways and gene expression (Hydbring et al, 2016; Pagano and Jackson, 2004). In this context, it could be speculated that cyclin D2 acts as an indispensable effector of sonic hedgehog signaling, which is required for the establishment and expansion of aNSCs and whose inactivation leads to a similar, but more severe, phenotype than the D2KO (Han et al, 2008; Li et al, 2013; Noguchi et al, 2019). How exactly cyclin D2 controls the formation of aNSCs remains to be determined. Previous reports indicate that D-cyclins, in addition to their role in proliferation, are also involved in fate specification and survival of various types of stem cells (Choi et al, 2014; Glickstein et al, 2007a; Glickstein et al, 2009; Lukaszewicz and Anderson, 2011; Pauklin et al 2016). However, we did not observe signs of increased cell death, neuron or astrocyte production in the early postnatal niches of mutant mice, indicating that cyclin D2 directs the cell divisions leading to the formation of aNSCs. Interestingly, the proportion of actively dividing NSCs in the SGZ of postnatal D2KO mice was unchanged, arguing against a general proliferation defect. This combined with the concomitant deficit in aNSC pool expansion suggests a requirement for cyclin D2 in fate specification during divisions producing aNSC, similar to what has been observed in the embryonic cortex (Glickstein et al, 2007a; Glickstein et al, 2009; Tsunekawa et al, 2012). Moreover, it is conceivable that cyclin D2 specifically promotes symmetric dNSC divisions required for the expansion of the aNSC pool, whereas cyclin D1 drives asymmetric divisions of dNSCs that foster developmental neurogenesis, consistent with a model proposed for embryonic radial

glia cells (Glickstein et al, 2009). To precisely determine the role of cyclin D2 in aNSC development, it will be necessary to generate mice that allow conditional deletion of cyclin D2 from NSCs as well as fate mapping of cyclin D2-expressing precursor cells.

The spatial origin of aNSCs remains a controversial topic. Whereas precursors from the DNe migrating along the DMS have traditionally been considered the source of aNSCs (Berg et al, 2019; Hodge et al, 2013; Li et al, 2009), others suggest the ventral sector of the hippocampus as origin of aNSCs (Li et al, 2013; Noguchi et al, 2019). Regardless of the embryonic origin, we aimed to determine where the final, cyclin D2-dependent mitoses that give rise to persistent aNSCs take place. Previous retroviral lineage tracing experiments in rats suggest that precursors residing in the hilar germinative matrix serve as a source of aNSCs (Namba et al, 2005). Moreover, studies in which pCAG-GFP was electroporated in the ventricular zone of mice imply that NSCs exiting the DNe later than E16 only generate neurons (Ito et al, 2014). Our fate mapping analyses using mCherry expressing γ-retroviral vectors show that aNSCs originate from precursor divisions inside the early postnatal DG, while precursors dividing at the same time in the DMS are almost exclusively neurogenic. Thus, besides uncovering a mechanism by which aNSCs are generated in the postnatal DG (cyclin D2-dependent mitosis), we show that the vast majority of aNSC precursors enter the DG before P2 and form a population separate from the exclusively neurogenic precursors that continue to migrate from the DNe into the postnatal DG.

These results challenge previous ideas that conceive aNSCs as mere remnants of dNSCs that simply change their behavior after colonizing the SGZ. Rather, they suggest that aNSCs are formed as a distinct population in a last wave of temporally and spatially restricted dNSC divisions that require cyclin D2. This conclusion finds support in studies showing that despite sharing common features with dNSC, aNSCs exhibit different cell division dynamics and a highly complex morphology, shift their mode of regulation from intrinsic to extrinsic, and differ in their transcriptomic landscape and marker expression (Borrett et al, 2022; Gebara et al, 2016; Hochgerner et al, 2018; Matsue et al, 2018; Nicola et al, 2015; Valcarcel-Martin et al, 2020). All this implies that adult hippocampal neurogenesis is distinct from the developmental process that gives rise to the bulk of the DG, which is further corroborated by recent work showing that the maturation and morphological properties of adult-born neurons differ from those that are neonatally born (Cole et al, 2020; Kerloch et al, 2019).

At this point it remains unclear if earlier precursors of aNSCs located outside the SGZ require cyclin D2 for their proliferation. Although our data show that primary neurogenesis constituting the DG during prenatal stages is largely independent of cyclin D2, it is possible that proliferation of perinatal hilar aNSC precursors

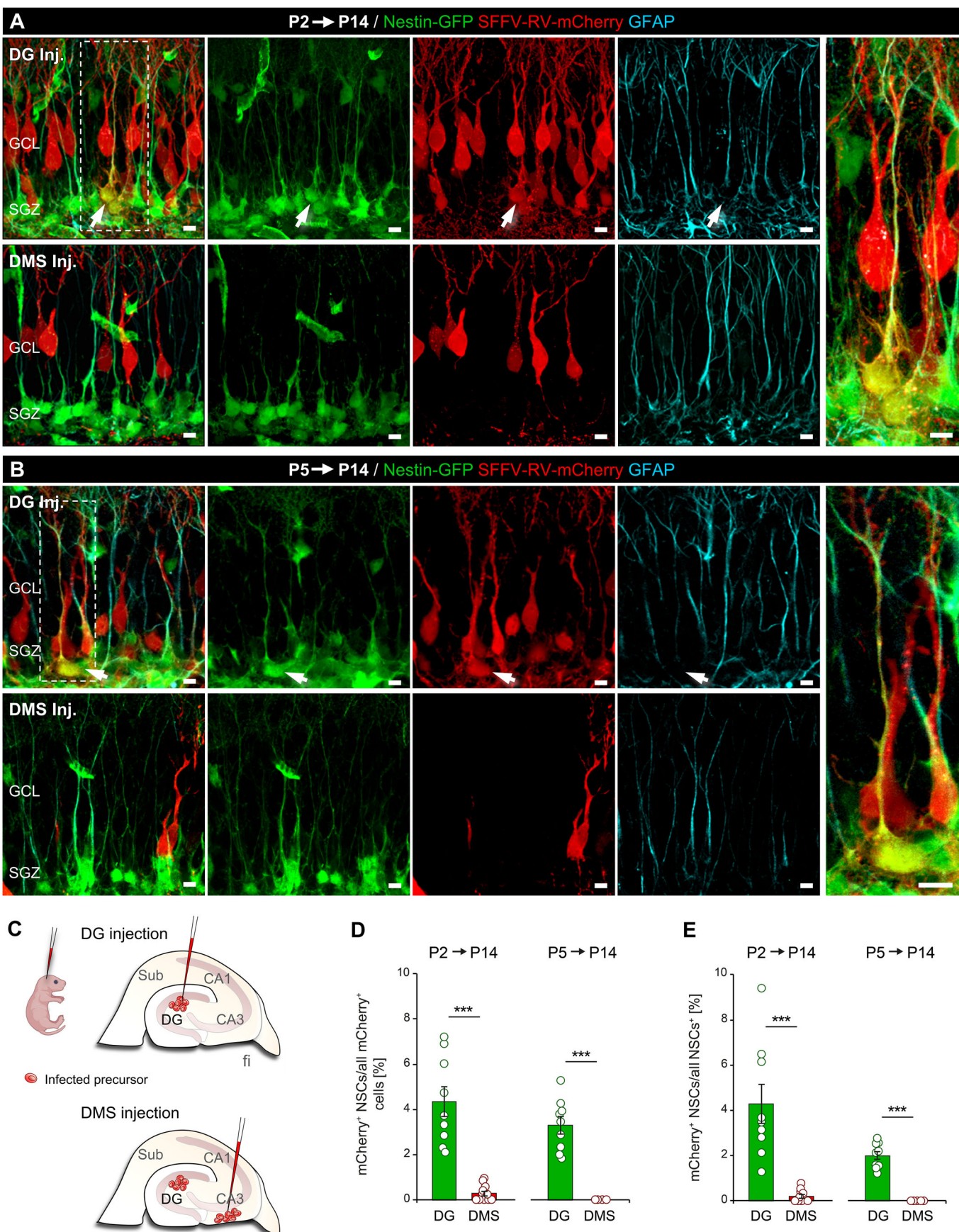

◄

**Figure 4. Adult NSCs are generated on-site in the early postnatal DG.**

(A, B) Confocal images of the nestin-GFP/GFAP-immunostained DG of P14 mice after SFFV-RV-mCherry injection into either the DG or the DMS at P2 (A) or P5 (B). DMS, dentate migratory stream. White arrows indicate mCherry⁺ NSCs. Scale bars 5 μm. (C) Schematic representation of the DG and DMS injection of retroviral vectors. (D) Percentage of NSCs among all transduced (mCherry⁺) cells (DG injections: $n = 9$ mice/age; DMS injections: $n = 12$ at P2, $n = 5$ at P5). (E) Percentage of mCherry⁺ NSCs among all NSCs (DG injections: $n = 9$ mice/age; DMS injections: $n = 12$ at P2, $n = 5$ at P5). Data information: All values represent mean ± SEM. Statistics: Welsh's $t$ test (D, E) and Student's $t$ test (E), ***$P < 0.001$. Source data are available online for this figure.

(Namba et al, 2005) or a subset of embryonic precursors in the preceding germinal niches relies on cyclin D2. At present, the identification of these cells is challenging due to the lack of appropriate markers or morphological features that allow reliable discrimination of different precursor populations, which is further complicated by the temporal changes in molecular marker composition in NSCs in the developing DG (Namba et al, 2005; Nicola et al, 2015). A potential contribution of hilar precursors to the pool of aNSCs, whose proliferation is also inhibited by the lack of cyclin D2, is compatible with our results.

Our work sheds light on the long-standing question about the formation of hippocampal aNSCs and identifies the critical period and spatial location in which they are born. Symmetric division (Bonaguidi et al, 2011) as well as time-dependent changes in cell division and self-renewal (Bottes et al, 2021; Harris et al, 2021; Ibrayeva et al, 2021) are able to extend the productive lifespan of the aNSC population. Still, the vast majority of the neurogenic output depends directly on the initial size of the aNSC population. An important prediction of this and previous findings (Ortega-Martinez and Trejo, 2015; Youssef et al, 2018) is that any pathophysiological event interfering with the establishment of aNSCs during this critical period will lead to lasting impairments of adult hippocampal neurogenesis and increased susceptibility to pathology.

The discovery of how and where aNSCs are generated in the rodent DG opens avenues for their manipulation and understanding of their cell nature in both rodents and humans. The existence of cyclin D2-dependent division leading to the formation of long-lasting precursors in the human DG remains unsolved. Further research is required to understand the progression of dNSCs in the human DG and whether the process is comparable to rodent DG development.

# Methods

## Reagents and tools

See Table 1.

## Methods and protocols

### Animals
All studies were carried out on cyclin D2 knockout/Nestin-GFP (D2KO) and wildtype/Nestin-GFP (WT) mice of mixed gender. The double-transgenic line was established by crossing heterozygous cyclin D2 knockout mice (Sicinski et al, 1996) with Nestin-GFP mice expressing enhanced green fluorescent protein (GFP) under control of rat nestin gene regulatory elements (Yamaguchi et al, 2000). The heterozygous ($ccnd2^{+/-}/GFP^+$) offspring of this breeding was mated to each other to obtain Nestin-GFP expressing

animals homozygous for the mutated (D2KO) or the wildtype (WT) $ccnD2$ allele. All mouse lines were maintained on a C57BL/6J background. Mice were kept under specific pathogen free conditions on a 14-h light/10-h dark cycle with food and water ad libitum. Animals were weaned at around P30, hence mice of all groups until P28 were housed as litters and/or with their mothers. Adult mice were group-housed. To minimize litter effects, littermates were assigned to groups in such a way that mice from at least three different litters contributed to a particular group, despite in the P7 WT group, which was recruited from only two litters. All animal procedures were in strict compliance with the European animal welfare regulations (EU directive 2010/63/EU and 2007/526/EC guidelines) and approved by the local authorities (Thueringer Landesamt für Verbraucherschutz, Bad Langensalza, Germany; University of the Basque Country (EHU/UPV) and the Comunidad Foral de Bizkaia Ethics Committees).

### Neurosphere assay
DGs of P7 and of 6–9 weeks old D2KO and WT mice were dissected and individually subjected to enzymatic and mechanical dissociation (Neural Tissue Dissociation Kit P, Miltenyi Biotec) as described earlier (Walker and Kempermann, 2014). The obtained single cells were suspended in 20 ml serum-free Neurobasal media (Gibco) containing 2% B27 (Gibco), 1× Glutamax (Gibco), 20 ng/ml EGF (PeproTech), 20 ng/ml bFGF (PeproTech), 50 U/ml Pen/Strep (Gibco), 20 μg/ml heparin (Sigma), and seeded into 96-well plates (200 μl/well, resulting densities: 2–10 cells/μl). Primary cultures were maintained at 5% $CO_2$ and 37 °C for 10 days. The total number and size of spheres that formed in each well were determined microscopically at ×200 magnification or with an IncuCyteZoom system (EssenBioscience, Germany). Only spheres larger than 20 μm in diameter were analyzed. To determine self-renewal capacity, we performed bulk culture serial passaging (10 × 7 days) as described earlier (Walker and Kempermann, 2014). Neurospheres of each animal were pooled, digested with Accutase (Sigma) and triturated into single cell suspensions. After counting cell numbers using a hemocytometer, cells were reseeded at a density of $1 \times 10^4$ cells/cm². The fold expansion relative to the seeded cell number was calculated and multiplied with the total of the previous passage to determine the theoretical total cell number.

### Injection of retroviral vectors
We adapted a method to perform intrahippocampal injections in accordance with our requirements: (1) the need to inject independently in two different regions (DG and DMS) of the hippocampus, and (2) its adaptation to the varying size of these regions during early postnatal development.

We traced the lineage of postnatal hippocampal progenitors with a γ-retroviral vector expressing the fluorescent protein mCherry under control of the spleen-focus forming virus promoter

**Table 1.  Reagents and tools.**

| Reagents/resource | Reference or source | Identifier or catalog number |
|---|---|---|
| Antibodies | | |
| Chicken anti-GFP | AvesLabs | Cat#GFP-1020; RRID: AB_10000240 |
| Goat anti-GFP | Acris | Cat#R1091P; RRID: AB_1002036 |
| Mouse anti-GFAP | Millipore | Cat#MAB360; RRID: AB_11212597 |
| Rabbit anti-GFAP | Synaptic Systems | Cat#173002; RRID: AB_887720 |
| Goat anti-GFAP | Abcam | Cat#ab53554; RRID: AB_880202 |
| Rabbit anti-S100b | DakoCytomation | Cat#Z0311; RRID: AB_10013383 |
| Rabbit anti-cyclin D2 | Santa Cruz | Cat#sc-593; RRID: AB_2070794 |
| Rabbit anti-cyclin D1 | Thermo Scientific | Cat#RM-9104-S1; RRID: AB_149913 |
| Rabbit anti-cyclin D1 | Abcam | Cat#ab16663; RRID: AB_443423 |
| Rabbit anti-Prox1 | ReliaTech | Cat#102-PA32S; RRID: AB_10013821 |
| Guinea pig anti-DCX | Millipore | Cat#AB2253; RRID: AB_1586992 |
| Mouse anti-MCM2 | BD Biosciences | Cat#610700; RRID: AB_2141952 |
| Rabbit anti-Ki67 | Novus Biologicals | Cat#NB110-89717 |
| Bacterial and virus strains | | |
| γ-Retroviral vector SFFV-RV-mCherry, pseudotyped with Eco envelope | Gomez-Nicola et al, 2014; PMID: 25531807 | N/A |
| Chemicals, peptides, and recombinant proteins | | |
| B27 | Gibco | Cat#17504-001 |
| Glutamax | Gibco | Cat#35050-038 |
| EGF | Peprotech | Cat#AF-100-15 |
| bFGF | Peprotech | Cat#100-18B |
| Heparin | Sigma | Cat#H3393 |
| Accutase | Sigma | Cat#A6964 |
| Vectastain ABC Elite Kit | Vector Laboratories | Cat#PK-6100 |
| Background Sniper | Biocare Medical | Cat#BS966L |
| Reveal Decloaker | Biocare Medical | Cat#RV1000M |
| Critical commercial assays | | |
| Neural Tissue Dissociation Kit P | Miltenyi Biotec | Cat#130-092-628 |
| Experimental models: organisms/strains | | |
| Mouse: C57BL/6J | The Jackson Laboratory | JAX: 000664 |
| Mouse: Cyclin D2 knockout | Sicinski et al, 1996 | N/A |
| Mouse: Nestin-GFP | Yamaguchi et al, 2000 | N/A |
| Mouse: Nestin-GFP | Mignone et al, 2004 | JAX:033927 |
| Oligonucleotides | | |
| Primer: GFP Forward: 5'-GCACGACTTCTTCAAGTCCGCCATGCC-3' | This study | N/A |
| Primer: GFP Reverse: 5'-GCGGATCTTGAAGTTCACCTTGATGCC-3' | This study | N/A |
| Primer: ccnd2 Primer D Reverse: 5'-GCTGGCCTCCAATTCTAATC-3' | Sicinski et al, 1996 | N/A |
| Primer: ccnd2 Primer G Forward: 5'-CCAGATTTCAGCTGCTTCTG-3' | Sicinski et al, 1996 | N/A |
| Primer: neo Primer N Forward: 5'-CTAGTGAGACGTGCTACTTC-3' | Sicinski et al, 1996 | N/A |

**Table 1.** (continued)

| Reagents/resource | Reference or source | Identifier or catalog number |
|---|---|---|
| Software and algorithms | | |
| SigmaPlot 14 | SigmaPlot | RRID: SCR_003210 |
| Fiji/Image J | Schneider et al, 2012; Schindelin et al, 2012 | RRID: SCR_002285, SCR_003070; http://fiji.sc |
| 3D-Sholl analysis | Ferreira et al, 2014 | http://fiji.sc/Sholl_Analysis |
| OriginPro 2019 | OriginLab Corporation | RRID: SCR_014212 |
| Adobe Illustrator | Adobe | RRID:SCR_010279 |
| MyCurveFit | | https://mycurvefit.com/ |
| Other | | |
| LSM710 confocal microscope | Carl Zeiss | N/A |
| LSM900 confocal microscope | Carl Zeiss | N/A |
| SP8 confocal microscope | Leica | N/A |
| IncuCyteZoom | EssenBioscience | N/A |
| 3DHistech panoramic digital slidescanner | Sysmex | N/A |

(SFFV-RV; Gomez-Nicola et al, 2014). The vector was based on RSF91.GFP.pre* (Schambach et al, 2006), pseudotyped with the envelope protein of mouse ecotropic MLV (Eco) and concentrated by centrifugation to a functional titer of $7.0 \times 10^9$/ml. Retroviral vectors rely on cell division to integrate their vector genome, since they can only gain access to the host cell's DNA after breakdown of the nuclear envelope during mitosis (Roe et al, 1993). This feature, which is sometimes perceived as a disadvantage, makes the vectors well suited to the identification of dividing cells.

P2 and P5 old Nestin-GFP mice (Mignone et al, 2004) were anesthetized by hypothermia by placing them in ice for 2 min and then secured to a platform placed in the stereotaxic apparatus. Aided by a fiber optic light, Lambda was localized by head transillumination and used as reference for the injections with a heat-pulled glass microcapillary at the following stereotaxic coordinates: for DG, −1 mm anteroposterior (AP), ±1.2 mm laterolateral (LL), −1.7 mm dorsoventral (DV); for the DMS −0.9 mm AP, ±1.4 mm LL, −1.7 mm DV. The microcapillary was introduced percutaneously to inject 0.3 μl of SFFV-RV-mCherry in each hippocampal region at a rate flow of 0.3 μl/min. After surgery, mice were quickly placed in a warm water bath and thereafter kept on a thermal blanket until their respiration, skin color and locomotor activity returned to normal. Finally, to facilitate re-acceptance by the mother, they were impregnated with a mixture of the mother's feces, bedding and water prior being returned to the breeding cage. The whole procedure for each pup was carried out in less than 10 min to minimize as much as possible maternal stress. No pups were rejected or abused by the dam. However, and despite mortality was almost negligible, all the pups that did not recover from anesthesia corresponded to the P5 age group, suggesting a major susceptibility to hypothermia at this age.

The implemented method allows the delivery of viral particles in the targeted anatomical regions, yet the injection accuracy is not 100% due to the variability in animal size and methodological limitations. Therefore, given the proximity of both injection sites and their small size at these postnatal stages, we employed a posteriori criteria to correctly assign injected animals to each group. To classify an injection as "DG," we ensured that no cells were labeled by the RV in the DMS. Furthermore, the path of the microcapillary was observable in several cases, assuring the identification of the injection site. To assign injections to the "DMS" group, transduced cells were required to populate the entire stream above the fimbria from the injection site into the DG. Injections that only reached the meninges could be distinguished from DMS injections due to their location forming a stream under the ventral blade of the DG and the characteristic elongated morphology of the cells in this region.

*Immunostaining*
Brains were obtained from different postnatal/juvenile ages (P0, P7, P10, P14, P28; at least 4 mice per group) as well as from adult mice (approx. P76; at least 4 mice per group; Fig. 2A). The P0 pups were decapitated and their brains fixed in 2% paraformaldehyde (PFA; w/v) in 0.1 M phosphate buffer (pH 7.4) at 4 °C for 20 h. All other mice were sacrificed by an overdose of isoflurane and transcardially perfused with ice-cold PBS (pH 7.4; 5 ml/min for 2 min) followed by 4% PFA in 0.1 M phosphate buffer, (pH 7.4; 5 ml/min for 8 min). After dissection, brains were removed urband post-fixed in PFA for 3 h at 4 °C. For long-term storage, the brains were cryoprotected consecutively in 10% and 30% sucrose (in PBS, 4 °C), cut midsagitally into the two hemispheres, frozen in 2-methylbutan (−25 to −30 °C) and stored at −80 °C. Hemispheres were sectioned either into 40 μm coronal sections on a sliding microtome (Epredia 400, Fisher Scientific, Germany) or into 50 μm sagittal sections on a Leica VT1200S vibratome (Leica Microsystems, Germany), and stored at −20 °C in antifreeze solution. For each antibody combination, every sixth hemisection was processed free-floating according to standard procedures described previously (Ansorg et al, 2015). To prevent batch effects, individual immunostaining rounds comprised samples from all groups, ideally one per group, to prevent batch effects arising from variations in staining quality. After rinsing with TBS (pH 7.4), sections were blocked in Background Sniper (Biocare Medical; for cyclin D1) or in TBSplus containing 0.2% Triton X-100, 3% or 10% donkey serum and 2%

BSA (1 h at room temperature), which was also used for antibody dilution. Primary antibodies were incubated for approximately 40 h at 4 °C followed by overnight incubation in secondary antibodies with three rinsing steps in-between. To stain nuclei, DAPI (Sigma) or Hoechst33342 (Invitrogen) were added at this step. Finally, sections were rinsed three times with TBS, mounted to gelatinized slides, air-dried and coverslipped with aqueous mounting medium (Fluoromount-G, Southern Biotech or DakoCytomation Fluorescent Mounting Medium, DakoCytomation). For staining with antibodies against cyclin D1 and MCM2, we performed an epitope retrieval before the first blocking step. Therefore, sections were either steamed for 5 min in Reveal (Biocare Medical; for cyclin D1) or in sodium citrate buffer (pH6; for MCM2), followed by rapid cooling on ice water and wash-out in TBS.

Immunoperoxidase staining resembled the standard protocol described above with the following modifications: Before blocking in TBSplus, sections were treated with 1.5% $H_2O_2$ for 30 min at RT. After primary antibody incubation, sections were sequentially incubated in biotinylated secondary antibody (all raised in donkey, Dianova; 1:500), avidin–biotin–peroxidase solution (Vectastain ABC Elite Kit, Vector Laboratories, Burlingame, CA, USA) for 1 h at room temperature, followed by 3,3'-diaminobenzidine (Sigma-Aldrich) signal detection and mounting in Neo-Mount (Merck-Millipore).

For mice injected with retroviral vectors, immunostaining was performed as previously described using the methods optimized for the use in transgenic mice (Encinas and Enikolopov, 2008; Encinas et al, 2011). Animals were transcardially perfused with 30 ml of PBS followed by 30 ml of 4% (w/v) PFA in PBS, pH 7.4. Next, the brains were removed and postfixed for 3 h at room temperature in the same fixative solution, then transferred to PBS-0.2% sodium azide and kept at 4 °C. Serial 70-µm-thick coronal sections were cut using a Leica VT1200S vibratome to properly visualize the extension of the SFFV-RV-mCherry transduction. For immunostaining, sections were incubated with blocking and permeabilization solution (PBSplus containing 0.25% Triton-X-100 and 3% BSA) for 3 h at room temperature, and then incubated overnight with the primary antibodies (diluted in PBSplus) at 4 °C. After the incubation, the primary antibody was removed and the sections were washed with PBS three times for 10 min. Next, the sections were incubated with fluorochrome-conjugated secondary antibodies diluted in PBSplus for 3 h at room temperature. After washing with PBS, the sections were mounted on gelatin coated slides and coverslipped with DakoCytomation Fluorescent Mounting Medium.

The following antibody combinations were used: chicken anti-GFP (AvesLabs, 1:1000), goat anti-GFP (Acris, 1:300–500), mouse anti-GFAP (Millipore, 1:1000), rabbit anti-GFAP (Synaptic Systems, 1:500), goat anti-GFAP (ABCAM, 1:500), rabbit anti-S100β (DakoCytomation, 1:500), rabbit anti-cyclin D2 (Santa Cruz, 1:25), rabbit anti cyclin D1 (Thermo Scientific, 1:50), rabbit anti cyclin D1 (ABCAM, 1:500), rabbit anti-Prox1 (ReliaTech, 1:1000), guinea pig anti-DCX (Millipore, 1:500), mouse anti-MCM2 (BD Biosciences, 1:250), rabbit anti-Ki67 (Vector Laboratories, 1:750), rabbit anti-DsRed (Abcam, 1:2000; for detecting mCherry), AlexaFluor-488 anti-chicken (Molecular probes, 1:500), AlexaFluor-488 anti-goat (Molecular probes, 1:500), RhX anti-guinea pig (Dianova, 1:500), RhX anti-mouse (Dianova, 1:500), RhX anti-rabbit (Dianova, 1:500), AlexaFluor647 anti-mouse (Dianova, 1:500), AlexaFluor647 anti-rabbit (Dianova, 1:500), AlexaFluor647 anti-goat (Dianova, 1:500).

### Image capture and analysis

All sections were imaged on Zeiss confocal microscopes (LSM710 and LSM900; Carl Zeiss, Jena, Germany) or on a Leica SP8 confocal microscope (Leica, Wetzlar, Germany). The signal from each fluorochrome was collected sequentially, except for DAPI and AlexaFluor647, which on the LSM900 were captured in parallel. Brightness, contrast, and background were adjusted equally for the entire image using supplier's software (Zeiss ZEN black edition and ZEN 3.0 blue edition; Leica LAS X Life Science). The total volume of each granule cell layer (GCL) was estimated using the ×10 objective of a Leica SP8 confocal microscope to completely visualize the DG in each section. The area tool of the LAS X software was used to measure the area of the GCL in each section. The corresponding section thickness was measured in 3 points in each slice using the z.stack setting. For the pictures of developing mouse DG with SFFV-RV-mCherry transductions, in which the DMS is illustrated, the 3DHistech panoramic digital slidescanner (Sysmex, Germany) was used.

Quantitative analysis of cell populations in situ was performed by design-based (assumption free, unbiased) stereology using a modified optical fractionator sampling scheme as previously described (Encinas and Enikolopov, 2008; Encinas et al, 2011). With a few exceptions, the group assignment of the animals was unknown to the person analyzing the data. Analyses were done plane-by-plane throughout the Z-stacks. For cell densities, quantifications were done maintaining the same Z-stack size between conditions, corrected for the physical section thickness (measured in the GCL) and normalized to one mm² SGZ. Total numbers of cells were estimated by multiplying the density with the area of the SGZ. Therefore, the length of the SGZ was determined in every slice of a single series, summed up per animal and multiplied by the section thickness (as cut), the section interval, and by two to obtain the bilateral SGZ area. The SGZ was distinguished from the GCL and hilus based on the following criteria: a tightly packed band of DAPI⁺ nuclei densely populated with GFP⁺ cell bodies, indicating the presence of NSCs and intermediate progenitors. At P0, where the SGZ is not yet established and dNSCs are distributed across the prospective SGZ and GCL, quantifications were done throughout the entire SGZ/GCL.

Confocal image stacks were captured along the entire rostro-caudal extend of the DG except the most caudal part (corresponding to Bregma ≤ −3.5 in adults), in which radial NSCs cannot be reliably identified by their morphology. Depending on the DG size along the rostro-caudal axis, one to five image stacks (512 × 512 pixels resolution) covering the SGZ/GCL/ML were random-systematically acquired per hemisection. In P0 mice, the entire DG was imaged. For quantifying NSCs, astrogliogenesis and apoptosis, a ×40/1.3 oil immersion objective was used to acquire image stacks of 100 µm × Y µm × 12.07 µm size (XYZ; 0.71 µm Z-interval). X was aligned in parallel to the SGZ and Y was variably adjusted to ensure that the SGZ, GCL and ML were completely covered. For classification of NSCs, optical fields covering the SGZ/GCL were obtained with a ×40 oil immersion objective. The hilus was imaged separately with the optical field size set to 10,000 µm². For evaluating the neuronally committed progenitor populations, 100 µm × 100 µm × 7 µm image stacks (Z-interval of 0.70 µm) covering the SGZ and GCL were captured using a ×63/1.4 oil objective. In case of Ki67 stainings, we imaged the entire DG with a ×40/1.2 glycerin immersion objective (Z-interval 0.59 µm).

NSCs were defined as GFP⁺GFAP⁺ cells with their soma in the SGZ and an apical process spanning towards the ML through

at least 2/3 of the GCL (Encinas and Enikolopov, 2008). For experiments during postnatal stages prior to P14, where many NSCs are still not morphologically adult-like (Brunne et al, 2013; Nicola et al, 2015), NSCs were quantified as such when they had their soma located inside the SGZ/GCL (P0) or SGZ (≥P7) and fulfilled the following criteria: Expression of GFP and GFAP and presence of a single radial process of variable length extending towards the ML. To identify proliferating NSCs, we used the marker Ki67. To classify NSC subtypes in adult mice, GFP$^+$GFAP$^+$ cells located in the SGZ (each GFP$^+$GFAP$^+$ cell in D2KO mice, approx. 50% of GFP$^+$GFAP$^+$ cells in WT mice) were categorized based on the expression of S100β and morphological criteria (Gebara et al, 2016; Martin-Suarez et al, 2019): α cells were defined as GFP$^+$GFAP$^+$S100β$^-$ cells with the soma located in the SGZ and a long radial process extensively branching at the border between GCL and ML. β cells were identified by their S100β$^+$ soma located in the SGZ or GCL and a short primary process branching more proximal and rarely extending into the ML. Ω cells displayed a multibranched morphology with several primary processes extending from a S100β$^-$ soma located in the GCL. For the morphological analysis of NSCs, individual GFP$^+$ NSCs whose cell bodies were located either in the SGZ or GCL were randomly selected (approx. 8–10 per animal) and imaged over their entire extend using a ×63 oil immersion objective (resolution of $1024 \times 1024$ pixels). The obtained Z-stacks were analyzed using the 3D-Sholl analysis plugin in Fiji (http://fiji.sc/Sholl_Analysis; Ferreira et al, 2014).

Neuronally committed type 2b progenitor cells were determined by their expression of GFP, Prox1 and DCX, while type 3 neuroblasts were identified as Prox1$^+$DCX$^+$ cells with an apical dendrite and occasionally weak GFP expression. Astrocytes were identified by their stellate morphology and the expression of GFAP, and further distinguished into immature and mature astrocytes based on the expression or lack of GFP. To provide spatial information about potential migration routes from the germinative matrices towards the ML, quantification was conducted separately in the outer half of the GCL (oGCL), in the inner and outer half of the ML (iML, oML), and the hilus (H). The SGZ, inner GCL and subpial region were excluded from analysis, because especially until P10 these regions were too densely packed with GFP$^+$GFAP$^+$ cells to clearly identify astrocytes.

The number of apoptotic cells was estimated from the same Z-stacks in which cyclin D1-positive cells were phenotyped. To increase sampling accuracy in the hilus, we additionally included image stacks acquired for analysis of cyclin D2 expression and neurogenesis (hence, approximately every second hemisection was analyzed). Apoptotic cells were identified by their densely stained and characteristically shaped nuclei, in which the aggregated chromatin appears as peripheral condensation or one solid, compact sphere (pyknosis) or fragmented into multiple spheres (karyorrhexis; Doonan and Cotter, 2008; Fig. EV3).

For the experiments using SFFV-RVs-mCherry injections, we quantified all transduced cells throughout the SGZ/GCL, following the same criteria as described before. We also quantified the number of NSCs in that area and assessed them for mCherry expression to obtain the percentage of NSCs transduced by the SFFV-RV.

Confocal images in the figures are displayed as maximum intensity Z-projections.

## Statistical analysis

Significance levels were assessed in Sigma Plot 14 (San Jose, CA, USA). Data were first tested for normality. If given, either a two-tailed Student's *t* test or a Welch's *t* test (in case of no homoscedasticity) were applied to compare a single dependent variable between two groups, otherwise we used a Mann–Whitney *U*-test. For comparing a single dependent variable between more than two groups, we applied a one-way ANOVA, while two-way ANOVAs were applied to assess the effect of two independent variables on one dependent variable. A one-way repeated measures (RM) ANOVA was used for comparing several dependent variables in one group. To examine the effect of two independent variables, of which one has been measured repeatedly, we applied a 2-RM ANOVA. All ANOVAs were followed by Tukey's post hoc tests to determine specific differences. A $P < 0.05$ was considered as statistically significant. Data represent mean ± SEM, sample sizes are indicated in the figure legends.

## Data availability

This study includes no data deposited in external repositories.

## Peer review information

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

## Acknowledgements

We thank the staff at the animal facilities of UPV/EHU and the University Hospital Jena, Laura Escobar at the Imaging Core Facility in Achucarro, and the technicians in the Jena Neurology department for technical support. We would also like to thank members of the Neural Stem Cell and Neurogenesis group in Leioa and of the Hippocampal Plasticity and Neurogenesis group in Jena for discussion and insight, as well as T. Lehmann for statistical advice and C. Schmeer for proofreading. OP-A received a UPV/EHU predoctoral fellowship. JME was supported by the Spanish Ministry of Economy and Competitiveness (MINECO; grants SAF2-015-70866-R, MINECO Ramón y Cajal Program: RYC 2012-11137 and MINECO PCIN-2016-128 (ERA-NET-NEURON III program)), by Basque Government (PIBA_2021_1_0018) and by Federación Española de Enfermedades Raras (FEDER). OWW received support from German Research Foundation (DFG; grants WI 830/12-1 and WI 830/12-2) and Marie Skłodowska-Curie Innovative Training Network (ITN; grant 859890 SmartAge). AU was supported by the Interdisciplinary Center for Clinical Research (IZKF; grant AMSP06). Some figures include icons from BioRender.com.

## Author contributions

**Oier Pastor-Alonso**: Conceptualization; Data curation; Formal analysis; Validation; Investigation; Visualization; Methodology; Writing—original draft; Writing—review and editing. **Anum Syeda Zahra**: Conceptualization; Data curation; Formal analysis; Validation; Investigation; Methodology; Writing—review and editing. **Bente Kaske**: Investigation. **Fernando Garcia-Moreno**: Conceptualization; Data curation; Supervision; Investigation; Methodology. **Felix Tetzlaff**: Investigation. **Enno Bockelmnn**: Formal analysis; Investigation; Methodology. **Vanessa Grunwald**: Investigation. **Soraya Martin-Suarez**: Formal analysis; Investigation. **Kristoffer Riecken**: Resources. **Otto Witte**: Resources; Supervision; Funding acquisition. **Juan Manuel Encinas**: Conceptualization; Resources; Data curation; Supervision; Funding acquisition; Investigation; Visualization; Writing—review and editing. **Anja Urbach**: Conceptualization; Resources; Data curation; Formal analysis; Supervision; Funding acquisition; Validation; Investigation; Visualization; Methodology; Writing—original draft; Project administration; Writing—review and editing.

## Funding

## Disclosure and competing interests statement

The authors declare no competing interests.

# Expanded View Figures

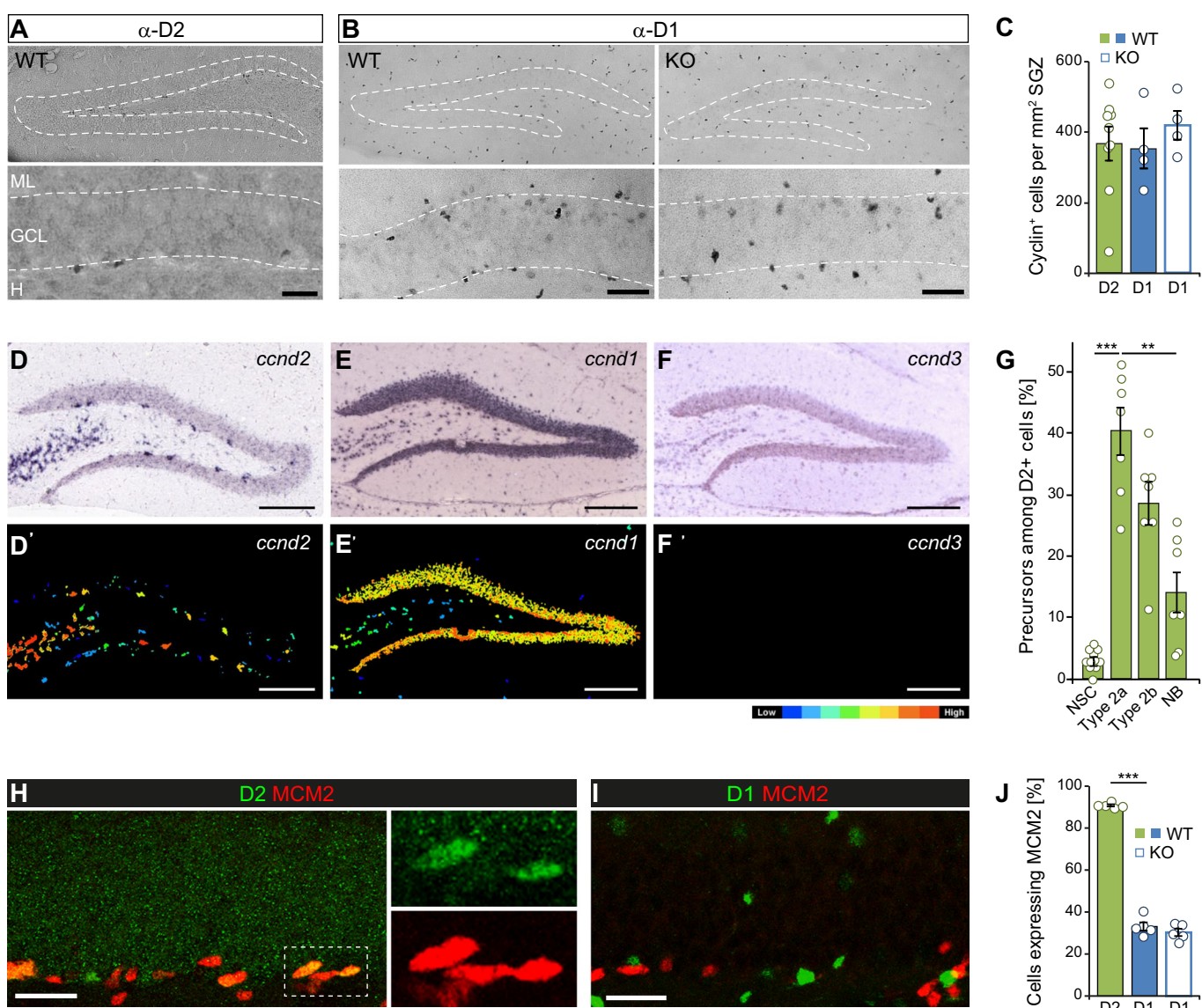

**Figure EV1.  Cyclin D2 is expressed by actively dividing progenitors in the adult SGZ.**

(A, B) Peroxidase staining of coronal brain sections illustrating the distribution of cyclin D2 (A) and D1 (B) in the DG of WT and D2KO mice. Upper panels: overview of the DG, lower panels: magnified image of the suprapyramidal blade. (C) Density of cyclin D$^+$ cells in the adult SGZ (D2: $n = 9$, D1: $n = 4$ mice/group). (D–F) Expression of *ccnd1* and *ccnd2* mRNA and lack of *ccnd3* in the adult DG. Allen Mouse Brain Atlas, https://mouse.brain-map.org/experiment/show/205, https://mouse.brain-map.org/experiment/show/69540507, https://mouse.brain-map.org/experiment/show/68191468. (D´–F´) Expression mask image display highlighting cells with highest probability of gene expression. (G) Quantification of the proportions of NSCs, type 2 cells and neuroblasts among cyclin D2$^+$ cells revealed a prevalence of transit-amplifying type 2 progenitors ($n = 7$ mice). (H, I) Confocal images illustrating the expression of MCM2 in cyclin D2$^+$ and D1$^+$ cell populations. Maximum intensity projections of 11 μm high Z-stacks. (J) Proportions of cyclin D2$^+$ and D1$^+$ cells expressing MCM2 ($n = 5$ mice/group). Data information: All values represent mean ± SEM. Statistics: One-way ANOVA (C, J) and One-way RM-ANOVA (G), **$P < 0.01$, ***$P < 0.001$; Scale bars: 25 μm (A, B, H, I) and 200 μm (D–F´). ML molecular layer, GCL granule cell layer, H hilus. Source data are available online for this figure.

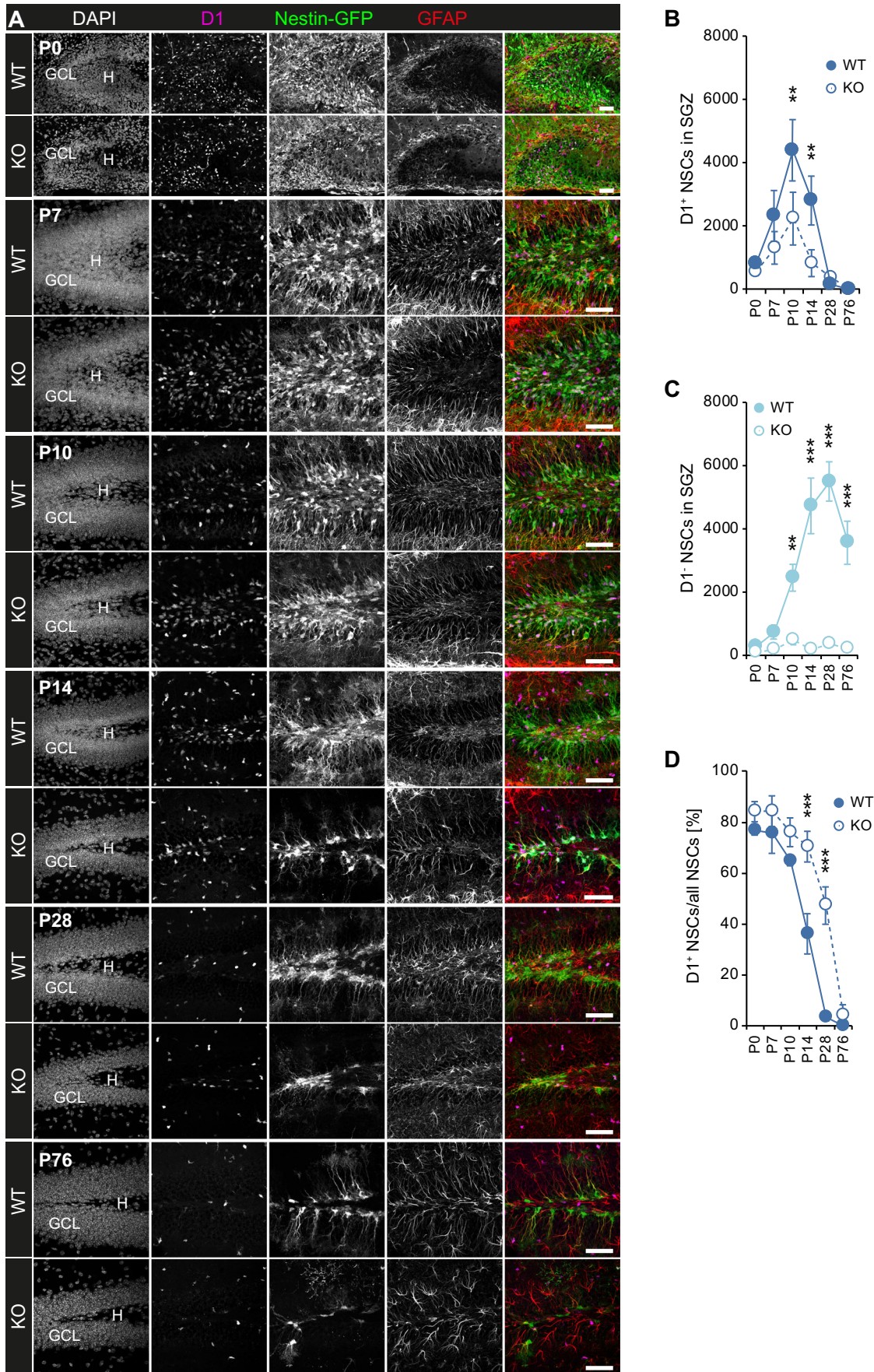

◀    **Figure EV2.   The lack of cyclin D2 impairs the postnatal expansion of the cyclin D1⁻ NSC population and diminishes the transient cyclin D1⁺ NSC population.**

(A) Confocal images of the developing DG of WT and D2KO mice immunostained against cyclin D1, nestin-GFP, GFAP and DAPI. Figures represent maximum intensity projections of 12.07 μm high *Z*-stacks. (B, C) Quantification of cyclin D1⁺ and D1⁻ NSC numbers in the SGZ. (B) Cyclin D1⁺ NSCs are a transient population. Deletion of cyclin D2 affects but does not prevent their appearance, suggesting that cyclin D1 may either partially compensate for the lack of cyclin D2 or designate a distinct dNSC population. (C) Deletion of cyclin D2 prevents the expansion of the cyclin D1⁻ NSC population. $N = 4$ mice/group except P10 KO and P14 WT with $n = 5$. (D) Proportions of NSCs expressing cyclin D1 are higher in mutant mice compared to WT ($n = 4$ mice/group except P10 KO and P14 WT with $n = 5$). Data information: Values represent mean ± SEM. statistics: 2-way ANOVA, $**P < 0.01$, $***P < 0.001$; Scale bars: 50 μm. H hilus, GCL granule cell layer.

                                                                  

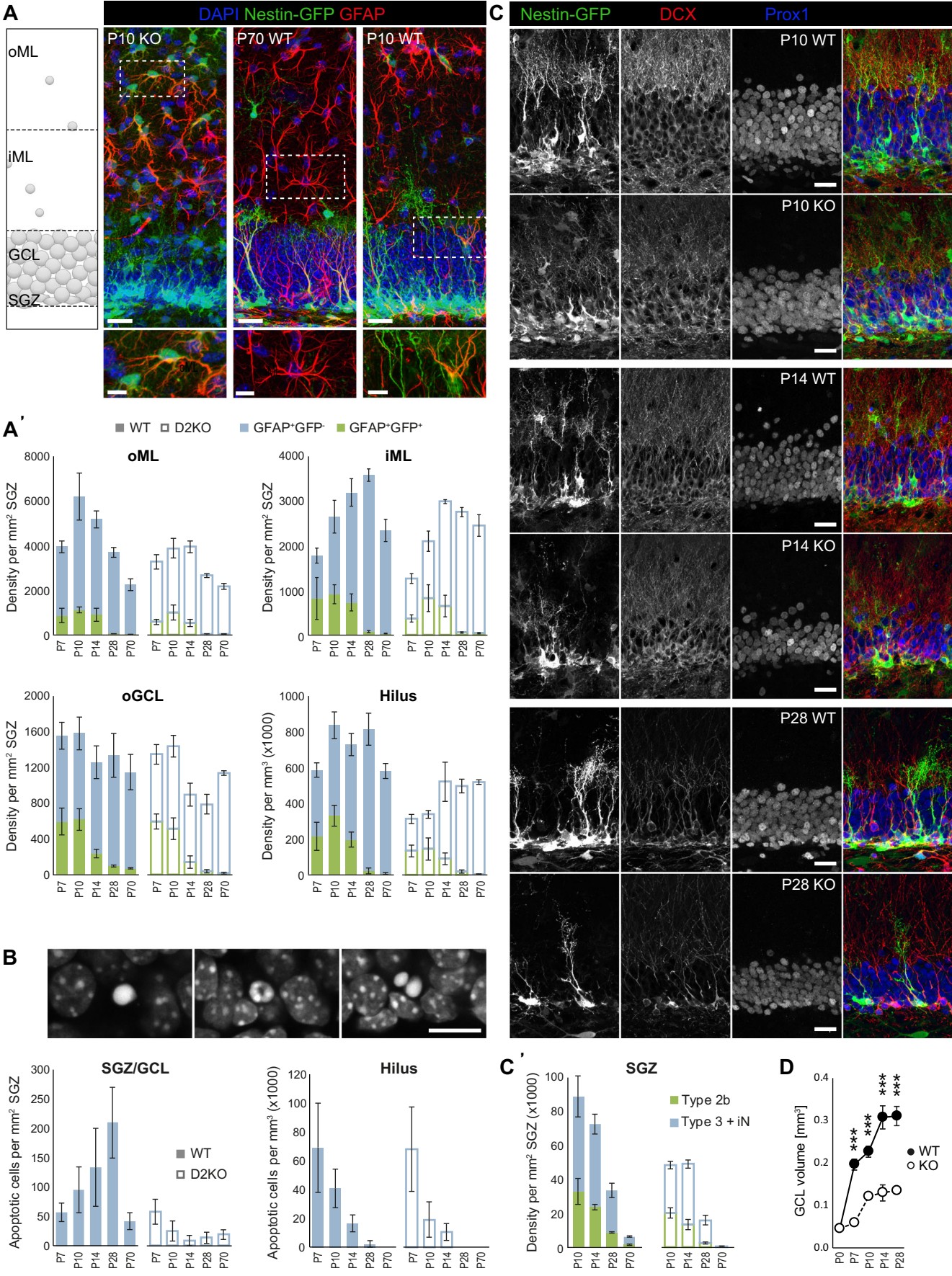

◀ **Figure EV3.** **The impaired formation of the aNSC pool in D2KO mice is not caused by apoptosis or differentiation of postnatal NSCs.**

(A) Postnatal astrogliogenesis is not increased upon deletion of cyclin D2. Top panel (from left to right): Sample confocal images of immature (GFAP$^+$GFP$^+$) and mature (GFAP$^+$GFP$^-$) stellate astrocytes in the ML and of an immature polar astrocyte in the GCL. Images represent maximum intensity projections of 12.07 μm confocal Z-stacks; scale bars 20 μm and 10 μm in magnified images). (A′) Quantification of immature and mature astrocytes ($n = 4$/group). Because of the extensive migration of dNSC-derived astrocytes in the postnatal DG (Brunne et al, 2010), analysis was performed in different layers of the DG. (B) Quantification of apoptotic cells ($n = 4$ mice/group except $n = 5$ in P10 and P14 WT). Top panel: DAPI staining showing the morphotypes of nuclei (pyknotic, donut-shaped and karyorrhectic; single optical planes) considered as apoptotic. Scale bar 10 μm. (C, C′) Quantification of neuroblasts and immature neurons in the postnatal SGZ ($n = 4$ mice/group). Images represent maximum intensity projections of 6.9 μm confocal Z-stacks, scale bars represent 20 μm. (D) Quantification of the GCL volume ($n = 5$ mice/group except $n = 6$ in P7 KO and $n = 4$ in P28 WT; 2-way ANOVA, ***$P < 0.001$). Data information: Values represent mean ± SEM. SGZ subgranular zone, GCL granule cell layer, iML inner half of the molecular layer, oML outer half of the ML representing the subpial germinative niche of the developing DG.

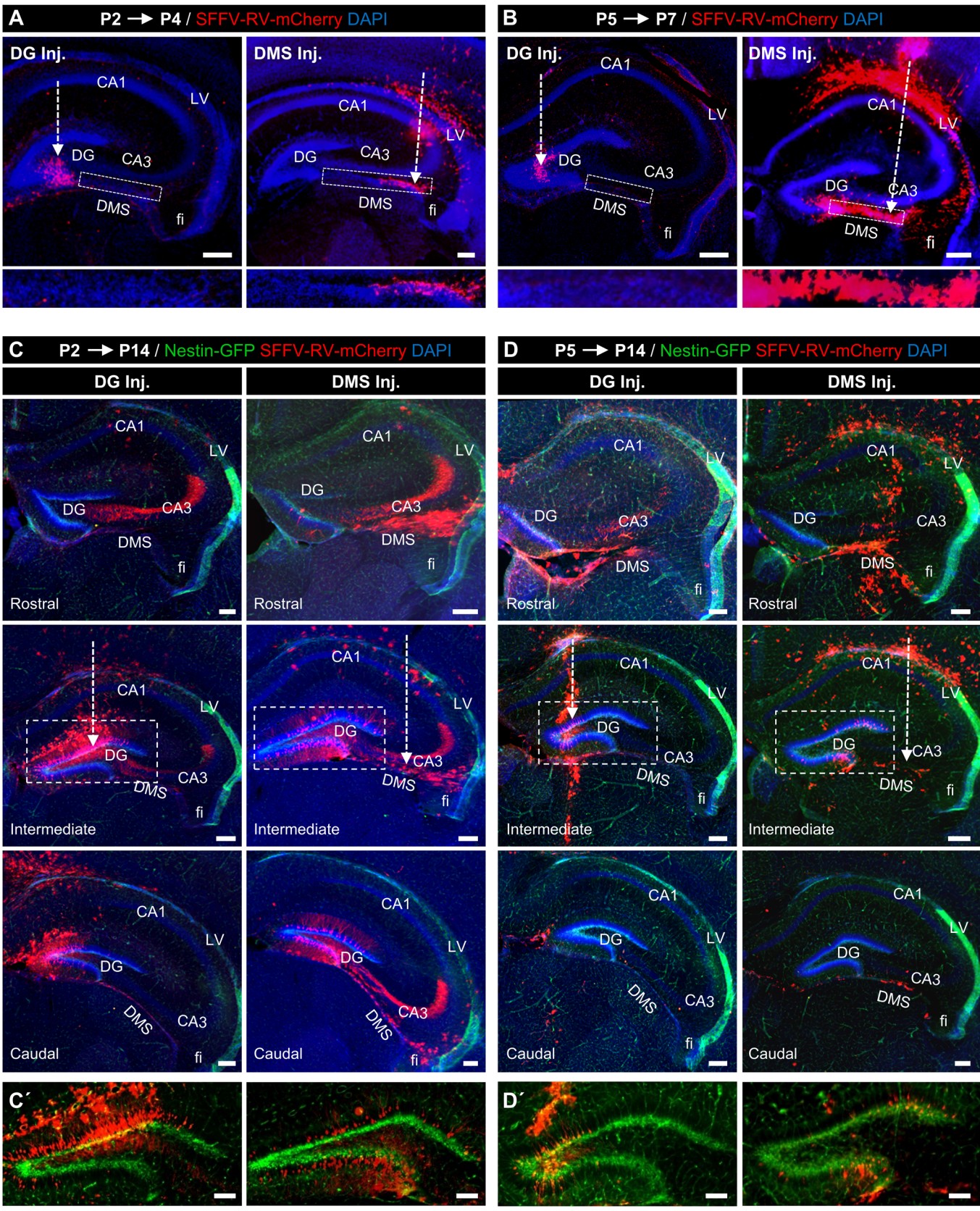

◀ **Figure EV4. Fluorescence slide-scanner images showing the spatial destination of SFFV-RV-mCherry-transduced cells after injection into the DG or in the DMS.**

(A, B) To evaluate the spatial precision of SFFV-RV injections into the DG or the DMS, C57Bl/6 mice were injected either at P2 (A) or P5 (B) and sacrificed 2 days later. The squares delimitate the DMS shown at higher magnification below, and arrows mark the injection path. At both ages, mCherry+ cells were observed exclusively at the injected site, verifying that the injection paradigm is effective for tracing the lineage of aNSCs from different postnatal niches. A wider distribution, including the DG near the fimbrodentate junction, was found in mice injected in the DMS, reflecting the migratory activity of the precursors located in that area. (C, D) Nestin-GFP expressing pups were injected at P2 (C) or P5 (D) and the spatial destination of transduced cells from rostral to caudal was assessed at P14. The arrows mark the injection path. Insets delineate the DG, which is shown at higher magnification in C′ and D′. In all cases, we observed mCherry+ cells located in the GCL, as well as axons entering the hilus. However, mCherry+ cells connecting the LV with the DG were observable only when the injection was done in the DMS. Data information: Scale bars = 200 μm. CA cornu ammonis, DG dentate gyrus, DMS dentate migratory stream, fi fimbria, GCL granule cell layer, LV lateral ventricle.

