## [Peer Review File · The EMBO Journal]

Generation of adult hippocampal neural stem cells occurs in the early postnatal dentate gyrus and depends on cyclin D2

Oier Pastor-Alonso, Anum Syeda Zahra, Bente Kaske, Fernando Garcia-Moreno, Felix Tetzlaff, Enno Bockelmann, Vanessa Grunwald, Soraya Martin-Suarez, Kristoffer Riecken, Otto Witte, Juan Encinas, and Anja Urbach

DOI: [10.15252/embj.2023113564](https://doi.org/10.15252/embj.2023113564)

Corresponding author(s): *Anja Urbach (anja.urbach@med.uni-jena.de)* , *Juan Encinas (jm.encinas@achucarro.org)*

Review Timeline:

Submission Date:	19th Jan 23
Editorial Decision:	15th Feb 23
Revision Received:	19th Sep 23
Editorial Decision:	16th Oct 23
Revision Received:	3rd Nov 23
Accepted:	20th Nov 23

Editors: Karin Dumstrei and Ioannis Papaioannou

Transaction Report:

Dear Anja,

Thank you for submitting your manuscript to The EMBO Journal. Your study has now been seen by three referees and their comments are provided below.

As you can see from the comments, the referees find the analysis interesting and suitable for The EMBO Journal. They raise several concerns that I would like to ask you to address in a revised version.

I think it would be helpful to discuss the raised points further and I am available to do so via email or video.

I thank you for the opportunity to consider your work for publication. I look forward to your revision.

with best wishes

Karin

Karin Dumstrei, PhD
Senior Editor
The EMBO Journal

We realize that it is difficult to revise to a specific deadline. In the interest of protecting the conceptual advance provided by the work, we recommend a revision within 3 months (16th May 2023). Please discuss the revision progress ahead of this time with the editor if you require more time to complete the revisions.

As a matter of policy, competing manuscripts published during this period will not negatively impact on our assessment of the conceptual advance presented by your study. However, we request that you contact the editor as soon as possible upon publication of any related work, to discuss how to proceed.

Use the link below to submit your revision:

Referee #1:

This is a well executed and well presented study that makes a useful contribution to our understanding of the origin of adult neural stem cells in the mouse dentate gyrus. There are two parts. The first addresses the role of CyclinD2 during the transition from developmental to adult neurogenesis in the first two weeks after birth with a detailed analysis of D2 KO mice. The conclusion is that CyclinD2 is required for the generation of adult NSCs while it is not required during the preceding phase of embryonic neurogenesis. The second part investigates where the divisions producing adult NSCs take place using a reporter retrovirus and concludes that adult NSCs are generated by divisions occurring in the DG rather than the dentate migratory stream. Overall the work is of excellent quality, the analysis is thorough and the findings are interpreted appropriately.

My only major concern is with the conclusion that adult NSCs represent a distinct population from developmental NSCs, which needs to be clarified. If adult NSCs are generated by divisions of precursors already present in the DG, it seems that these dividing precursors could be developmental NSCs, producing adult NSCs that switch in their requirement for CyclinD2 (which could happen within the same lineage). If not, what are these precursors and how do they differ from dNSCs? The discussion should clarify in what respect aNSCs are distinct from dNSCs and what is the evidence that aNSCs do not originate from divisions of dNSCs. The conclusion that adult NSCs represent a distinct population from developmental NSCs should be better supported or toned down in the abstract, results and discussion sections.

Along the same line, a paragraph of the Discussion (p. 16) contains the two following statements that seem somewhat contradictory: "These results challenge previous ideas that conceive aNSCs as remnants of embryonic DG development" and "Our results are not at odds with aNSCs being an endpoint of one developmental lineage". If there is a difference between "endpoint of a developmental lineage" and "remnant of embryonic DG development", it needs to be better explained.

Minor points that should be addressed:

In Fig. 1A, what is the marker used to label NSCs?

You should mention in the Results or Discussion that the conclusion that aNSCs are generated by divisions taking place in the DG is based on the fact that retroviruses infect only dividing cells.

Referee #2:

In their study, Pastor et al investigate the role of cyclin D2 in the generation of adult hippocampal neural stem cells (NSCs) during postnatal development. The authors address two unresolved and important questions: 1. While NSCs in the adult brain differ in many way from those in development, it is unknown how the transition from developmental to adult NSCs is mediated. 2. It is still controversially debated where adult NSCs spatially originate from. To address those questions, Pastor et al use a quite simple genetic approach by combining a constitutive knock out of cyclin D2 crossed to Nestin-GFP reporter mice to visualize NSCs. In a carefully conducted and analyzed experimental approach, the authors show that in adult stages, the number of adult NSCs is dramatically reduced in D2KO mice and the residual adult NSCs are mostly quiescent. Constitutive deletion of cyclin D2 prevents the formation of adult NSCs early on, while DG development and proliferation of developmental NSCs in the perinatal DG is not affected (potentially due to sufficient D1 function). The loss of aNSCs is neither a pure proliferation defect (residual NSCs proliferate at normal rates), nor caused by transdifferentiation or cell death (as assessed by quantifications of astrogenesis, neurogenesis and apoptosis). Instead, the deletion of D2 leads to the loss of cell divisions, which lead to the formation of adult NSCs. To answer the second question, the authors use retroviral labeling at different locations and developmental times to show that the origin of adult NSCs are precursors within the DG and not the DMS (dorsal migratory stream). They propose that these two lineages separate before P2 (the time in which the earliest RV-labeling took place). This manuscript contains important answers to important questions and it was a pleasure to read it. The study is straightforward, and the figures and images presented are very clear. However, the simple genetic model used does not allow to conclusively state whether or not these data support a "set-aside" or the "continuous" model of neurogenesis. Here, I would be more careful in making such general conclusions. The identification of potential precursors for adult NSCs is an important step forward to

delineate the question raised above (set-aside or continuous). As mentioned in the discussion, targeting specifically these progenitors for fate-mapping will reveal if they are really a separate trait of NSCs different from developmental ones. And conditional deletion of D2 specifically in those adult NSC precursors will ultimately reveal the role of D2 in the generation of adult NSCs. Overall, I consider the manuscript to be adequate and appropriate for publication in the EMBO Journal. Very few and only minor comments should be addressed further:

- The Nestin-GFP line is a transgenic mouse and the author genotype against GFP. Does their genotyping protocol or breeding strategy allow to discriminate between homozygous and heterozygous GFP? And which mice did they use to compare D2KO to D2WT - homozygous GFP or heterozygous GFP or both?

- In the synopsis it is unclear to me what the third color of granule neurons is representing. Adult-born are green, developmental-born are blue and then there is a third population in grey?

Referee #3:

This manuscript entitled "Generation of adult hippocampal neural stem cells occurs in the early postnatal dentate gyrus and depends on cyclin D2" investigates the role of cyclin D2 in the developmental generation of adult neural stem cells in the mouse hippocampus. First, the manuscript characterizes the number and morphology of neural stem cells in D2 knock out (KO) mice, which have previously been described to have low levels of adult neurogenesis and smaller dentate gyrus regions (Ansorg, 2012). Fewer neural stem cells are found in D2 KO dentate gyrus. Then, cyclin D1 and cyclin D2 expression in neural stem cells was assessed in wildtype dentate gyrus across early postnatal time points. While expression of D1 and D2 in the neural stem cell population gradually decreased across postnatal development (similar to overall proliferation levels), there was a transient expression of D2 in the subgranular zone (SGZ) region from P7-P14. While wildtype mice exhibited an expansion of the NSC population between P7-P14 in the SGZ, Cyclin D2 KO prevented much of this expansion, such that the number of NSCs and the SGZ area were reduced in adult mice. Finally, retroviral tracing in the DG proper or the dentate migratory stream at P2 or P5 showed that most adult NSCs are generated from precursors in the DG proper, rather than in the DMS.

Overall the experiments in the manuscript aim to address interesting questions about dentate gyrus development and the generation of adult neural stem cells during development, two topics that are not well studied. Clearly, this and previous work show that cyclin D2 is important for postnatal development of the dentate gyrus. However, the early postnatal dentate gyrus is still being formed and many of the quantification methods used in this paper are adopted from the adult and do not make sense in the developmental context. For example, the SGZ is not formed at P0 or P7, yet many quantifications were made "per SGZ". This could bias the interpretation of the data. In addition, multiple conclusions in the text overstate the data and should be modified to reflect the data more accurately. Below are comments that could be addressed to strengthen the manuscript.

Major Comments:

1. The Methods state that: "For experiments during postnatal stages prior to P10, where NSCs are still not morphologically adult-like (Brunner et al., 2013; Nicola et al., 2015), NSCs were quantified as such when they had their soma located inside the SGZ/GCL and fulfilled the following criteria: Expression of GFP and GFAP and presence of a radial process extending towards the ML." The following questions are related to quantification in Fig. 3. (1) There are a large number of Nestin-GFP+ cells that are located in the hilar area at early ages (P0 - P10/14). Were GFP+/GFAP+ cells in the hilar area excluded from the analysis at these ages? Might this cause a bias, especially at P0 when the dentate gyrus morphology is so immature and unformed? (2) GFAP does not look like a very good marker at P0 (Fig. 2B). Might another marker, like Sox2, be better for quantifying neural stem cells across ages?

2. Fig. 2D and Fig. 3D quantify "NSCs in SGZ". How do you define the SGZ at time points before P14 (Fig. 2D), especially P0 and P7? Fig. 2D suggests that virtually no NSCs (D2+ or D2-) are in the defined "SGZ" at P0, despite the fact that most NSCs at P0 are D2+ (Fig. 2C). Then are the P0 NSCs all outside of the defined "SGZ"? The definition of "SGZ" at P0 and P7 could be critically important for the conclusion that "D2+ NSCs are a transient population that appears shortly after birth and increases significantly until P10, before declining and returning to nearly P0 levels at P28 (Figure 2D)." It might be more comprehensive to count NSCs in the entire DG region.

3. The manuscript claims that most NSCs are D2+ before P14 (Fig. 2C), but that a D2+ NSC population is only transiently present in the SGZ during early postnatal development (Fig. 2D). It is possible that at P0 and P7 there are few D2+ NSC in the SGZ because the SGZ has not fully formed. Then it is possible that the decline of D2+ NSCs from P10 to P28 is simply the normal age-dependent decline in actively dividing NSCs (Fig. 3B).

4. Is the number of Ki67+ NSCs at P0 different between WT and KO mice? Do you have the data for the P0 time point in Fig. 3C? It would be best if Fig. 3C could mirror Fig. 3B.

5. "To examine whether the lack of aNSCs in the mutant DG was indeed caused by a deficit in their generation instead of premature differentiation or death of dNSCs, we quantified the numbers of astrocytes, neuroblasts and apoptotic cells in the

different layers of the developing DG (Figure S3)." This statement is misleading because the density was quantified, not the total numbers. The total numbers of these cell types are much lower in KO because the SGZ surface area is decreased (Fig. 3F).

6. Fig. 2B Nestin-GFP staining is incomparable to Fig. 3 Nestin-GFP staining. Staining in Fig. 2B seems like it would be difficult to quantify. Are there better images for Fig. 2B for at least P0 and maybe also P7?

7. "These data demonstrate that both D-cyclins are highly expressed by dNSCs during the time of peak proliferation in the developing DG (Altman and Bayer, 1990a; Bond et al., 2020) and are then downregulated making the NSCs transition to a quiescent adult state (Berg et al., 2019)." The data shown suggest an association between D-cyclin expression in NSCs and the timing of the NSC transition to a quiescent adult state, but they do not provide evidence that the change in D-cyclin expression causes ("makes") the NSCs transition.

8. "Strikingly, mCherry+ aNSCs were only found when RV-SFFVmCherry was injected into the DG but not into the DMS (Figure 4D-E). This result indicates that aNSCs are generated exclusively from precursors located in the DG early after birth, while precursors dividing in the postnatal DMS do not contribute to the formation of aNSCs and adult neurogenesis." To say that adult NSCs are generated exclusively from precursors in the DG early after birth is a bit of an overstatement of the data. The data (Fig. 4) certainly provide evidence that more aNSCs were generated from DG precursors than DMS precursors, but there are still a small number of aNSCs generated from P2 DMS precursors.

9. Fate mapping could provide more robust evidence that D2 is required for the generation of adult NSCs. The SFFV-RV-mCherry virus could be injected into the dentate gyrus of wildtype and D2 KO mice at P2 just as in Fig. 4, similar quantifications to Fig. 4D-E could be performed. The hypothesis would be that fewer NSCs would be generated in D2 KO mice.

Dear reviewers,

We thank all reviewers for their positive feedback and the valuable comments that helped us to improve our manuscript. We revised it accordingly and highlighted all the changes to facilitate traceability. Each comment has been addressed as you will see in our point by point responses below.

Referee #1:

This is a well executed and well presented study that makes a useful contribution to our understanding of the origin of adult neural stem cells in the mouse dentate gyrus. There are two parts. The first addresses the role of CyclinD2 during the transition from developmental to adult neurogenesis in the first two weeks after birth with a detailed analysis of D2 KO mice. The conclusion is that CyclinD2 is required for the generation of adult NSCs while it is not required during the preceding phase of embryonic neurogenesis. The second part investigates where the divisions producing adult NSCs take place using a reporter retrovirus and concludes that adult NSCs are generated by divisions occurring in the DG rather than the dentate migratory stream. Overall the work is of excellent quality, the analysis is thorough and the findings are interpreted appropriately.

Response: We thank the reviewer for the positive feedback and the constructive comments that helped to improve our manuscript.

My only major concern is with the conclusion that adult NSCs represent a distinct population from developmental NSCs, which needs to be clarified. If adult NSCs are generated by divisions of precursors already present in the DG, it seems that these dividing precursors could be developmental NSCs, producing adult NSCs that switch in their requirement for CyclinD2 (which could happen within the same lineage). If not, what are these precursors and how do they differ from dNSCs? The discussion should clarify in what respect aNSCs are distinct from dNSCs and what is the evidence that aNSCs do not originate from divisions of dNSCs. The conclusion that adult NSCs represent a distinct population from developmental NSCs should be better supported or toned down in the abstract, results and discussion sections.

Response: Thank you for this comment that has helped us to express our ideas more clearly. In fact, by saying that aNSCs are distinct from dNSCs, we did not mean to claim that they are separate lineages, i.e. that aNSCs do not arise from dNSCs. This could not be assessed with our experimental approach and we found no evidence for it.

The distinction between aNSCs and dNSCs is based on our findings that the precursors of aNSCs are restricted to the early postnatal DG and differ from the dNSCs that establish the DG prenatally and those still coming via the DMS (RV-analysis) in their requirement for D2 to generate progeny and in their potential to generate aNSCs. It is further supported by previous work showing 1) expression of LPAR1 in NSCs from P14 onwards (Valcarcel-Martin *et al.*, 2020) but not before, 2) distinct transcriptomic signatures of NSCs at early postnatal stages compared to NSCs at juvenile and adult stages (Hochgerner *et al.*, 2018), 3) morphological and division-related changes (high-rate of proliferation *versus* quiescence) from P14 onwards (Hochgerner *et al.*, 2018; Nicola *et al.*, 2015; Berg *et al.*, 2019; our data), and 4) the complex morphology of aNSCs indicating a high degree of differentiation (Gebara *et al.*, 2016).

We have toned down the conclusion in the abstract (page 3, lines 41 ff.), introduced the concept better in the introduction (page 5, lines 89 ff.) and discuss the evidence for aNSCs being distinct from dNSCs more extensively in the discussion section (page 13, line 266, page 16, lines 271 ff.).

Along the same line, a paragraph of the Discussion (p. 16) contains the two following statements that seem somewhat contradictory: "These results challenge previous ideas that conceive aNSCs as remnants of embryonic DG development" and "Our results are not at odds with aNSCs being an endpoint of one developmental lineage". If there is a difference between "endpoint of a developmental lineage" and "remnant of embryonic DG development", it needs to be better explained.

Response: Thank you for pointing this out. By "remnants" we mean dNSCs that contributed to the structural development of the DG and that merely change their behavior ("transform" into aNSCs) after colonizing the SGZ. Our data show that this is not the case. Rather, aNSCs are formed in a last wave of temporally and spatially restricted dNSC divisions that are strictly D2-dependent, unlike the preceding dNSC divisions. Both scenarios are compatible with aNSCs being the endpoint of a developmental lineage that they share with dNSCs.

We revised the discussion which now explains our concept of "remnants" and the evidence for aNSCs being distinct from dNSCs in more detail (page 16, lines 271 ff.).

Minor points that should be addressed:

In Fig. 1A, what is the marker used to label NSCs?

Response: The figure shows a DAB staining of GFP. We rephrased the figure legend to make this clearer: „ABC peroxidase staining of GFP in adult WT and D2KO mice illustrating the lack of nestin-expressing aNSCs in D2KO mice.“

You should mention in the Results or Discussion that the conclusion that aNSCs are generated by divisions taking place in the DG is based on the fact that retroviruses infect only dividing cells.

Response: We thank the referee for this comment. He/she is absolutely right, that this information is very relevant to some of the readers and indeed was missing. In the revised version, we briefly introduce the concept in the results (page 12, line 248 ff.) and explain it in more detail in the methods (page 21, line 418 ff.).

Referee #2:

In their study, Pastor et al investigate the role of cyclin D2 in the generation of adult hippocampal neural stem cells (NSCs) during postnatal development. The authors address two unresolved and important questions: 1. While NSCs in the adult brain differ in many way from those in development, it is unknown how the transition from developmental to adult NSCs is mediated. 2. It is still controversially debated where adult NSCs spatially originate from. To address those questions, Pastor et al use a quite simple genetic approach by combining a constitutive knock out of cyclin D2 crossed to Nestin-GFP reporter mice to visualize NSCs. In a carefully conducted and analyzed experimental approach, the authors show that in adult stages, the number of adult NSCs is dramatically reduced in D2KO mice and the residual adult NSCs are

mostly quiescent. Constitutive deletion of cyclin D2 prevents the formation of adult NSCs early on, while DG development and proliferation of developmental NSCs in the perinatal DG is not affected (potentially due to sufficient D1 function). The loss of aNSCs is neither a pure proliferation defect (residual NSCs proliferate at normal rates), nor caused by transdifferentiation or cell death (as assessed by quantifications of astrogenesis, neurogenesis and apoptosis). Instead, the deletion of D2 leads to the loss of cell divisions, which lead to the formation of adult NSCs. To answer the second question, the authors use retroviral labeling at different locations and developmental times to show that the origin of adult NSCs are precursors within the DG and not the DMS (dorsal migratory stream). They propose that these two lineages separate before P2 (the time in which the earliest RV-labeling took place).

This manuscript contains important answers to important questions and it was a pleasure to read it. The study is straightforward, and the figures and images presented are very clear.

However, the simple genetic model used does not allow to conclusively state whether or not these data support a "set-aside" or the "continuous" model of neurogenesis. Here, I would be more careful in making such general conclusions. The identification of potential precursors for adult NSCs is an important step forward to delineate the question raised above (set-aside or continuous). As mentioned in the discussion, targeting specifically these progenitors for fate-mapping will reveal if they are really a separate trait of NSCs different from developmental ones. And conditional deletion of D2 specifically in those adult NSC precursors will ultimately reveal the role of D2 in the generation of adult NSCs. Overall, I consider the manuscript to be adequate and appropriate for publication in the EMBO Journal. Very few and only minor comments should be addressed further:

Response: We appreciate your positive feedback and are pleased that you found our manuscript clear and straightforward.

We have revised the abstract and discussion and toned down our conclusions (page 3, lines 41 ff.; page 16). The reviewer is correct that the experimental approach used in our study does not allow to clarify the model of neurogenesis. We think this discussion is relevant for the field, but in agreement with the reviewer's comment we cannot address it conclusively in this manuscript. Therefore, to avoid any confusion we deleted this part of the discussion.

- The Nestin-GFP line is a transgenic mouse and the author genotype against GFP. Does their genotyping protocol or breeding strategy allow to discriminate between homozygous and heterozygous GFP? And which mice did they use to compare D2KO to D2WT - homozygous GFP or heterozygous GFP or both?

Response: We thank the reviewer for this comment. Since the Nestin-GFP line used in this study has been generated by pronuclear injection, genotyping does not allow for discrimination between homozygous and heterozygous GFP alleles or quantification of copy number. According to our breeding scheme, it can be assumed that both homo- and heterozygous GFP mice were included. However, it is highly unlikely that the number of GFP alleles explains the D2KO effects observed in the present study, since the GFP-expressing D2KO and WT mice were littermates, and homo- and heterozygosity for GFP should display a balanced distribution between them and not cause bias in the data.

- In the synopsis it is unclear to me what the third color of granule neurons is representing. Adult-born are green, developmental-born are blue and then there is a third population in grey?

Response: We regret the lack of clarity in the last part of our synopsis. In principle, the gray population represented all granule neurons that are not the direct descendants of the blue and green precursor cells shown on the left, i.e. those originating from any other dNSC or aNSC. To avoid any ambiguity, we colored the grey cells green and blue according to their location and putative origin.

Referee #3:

This manuscript entitled "Generation of adult hippocampal neural stem cells occurs in the early postnatal dentate gyrus and depends on cyclin D2" investigates the role of cyclin D2 in the developmental generation of adult neural stem cells in the mouse hippocampus. First, the manuscript characterizes the number and morphology of neural stem cells in D2 knock out (KO) mice, which have previously been described to have low levels of adult neurogenesis and smaller dentate gyrus regions (Ansorg, 20212). Fewer neural stem cells are found in D2 KO dentate gyrus. Then, cyclin D1 and cyclin D2 expression in neural stem cells was assessed in wildtype dentate gyrus across early postnatal time points. While expression of D1 and D2 in the neural stem cell population gradually decreased across postnatal development (similar to overall proliferation levels), there was a transient expression of D2 in the subgranular zone (SGZ) region from P7-P14. While wildtype mice exhibited an expansion of the NSC population between P7-P14 in the SGZ, Cyclin D2 KO prevented much of this expansion, such that the number of NSCs and the SGZ area were reduced in adult mice. Finally, retroviral tracing in the DG proper or the dentate migratory stream at P2 or P5 showed that most adult NSCs are generated from precursors in the DG proper, rather than in the DMS.

Overall the experiments in the manuscript aim to address interesting questions about dentate gyrus development and the generation of adult neural stem cells during development, two topics that are not well studied. Clearly, this and previous work show that cyclin D2 is important for postnatal development of the dentate gyrus. However, the early postnatal dentate gyrus is still being formed and many of the quantification methods used in this paper are adopted from the adult and do not make sense in the developmental context. For example, the SGZ is not formed at P0 or P7, yet many quantifications were made "per SGZ". This could bias the interpretation of the data. In addition, multiple conclusions in the text overstate the data and should be modified to reflect the data more accurately. Below are comments that could be addressed to strengthen the manuscript.

Response: We thank the reviewer for the careful evaluation and the constructive comments that we addressed in the revised manuscript and our point-by-point responses below.

Major Comments:

1. The Methods state that: "For experiments during postnatal stages prior to P10, where NSCs are still not morphologically adult-like (Brunner et al., 2013; Nicola et al., 2015), NSCs were quantified as such when they had their soma located inside the SGZ/GCL and fulfilled the following criteria: Expression of GFP and GFAP and presence of a radial process extending towards the ML." The following questions are

related to quantification in Fig. 3. (1) There are a large number of Nestin-GFP+ cells that are located in the hilar area at early ages (P0 - P10/14). Were GFP+/GFAP+ cells in the hilar area excluded from the analysis at these ages? Might this cause a bias, especially at P0 when the dentate gyrus morphology is so immature and unformed? (2) GFAP does not look like a very good marker at P0 (Fig. 2B). Might another marker, like Sox2, be better for quantifying neural stem cells across ages?

Response: The reviewer is right that the perinatal hilar region harbors large numbers of nestin-expressing precursor cells, a portion of which correspond to dNSCs that give rise to granule neurons and potentially aNSCs. We are convinced that a detailed analysis of this cell population would provide interesting and important answers about the dNSC dynamics in the developing DG. However, peri- and early postnatal hilar GFP+/GFAP+ cells are heterogeneous, comprising astrocyte progenitors and putative dNSCs that cannot be clearly differentiated morphologically. Thus, GFP+/GFAP+ cell quantifications in the hilus would be inconclusive. We based our study on the SGZ or prospective SGZ to restrict our conclusions to the aNSC population that will be established there.

We also agree that the sole use of GFAP is not sufficient for identifying NSCs. Unfortunately, there is no specific marker for NSCs, so their identification is always based on combinations of different marker proteins and morphological criteria, as have been applied in the present study. Using a different marker like Sox2 would not help either. Sox2 is even less specific for NSCs than GFAP, because it is expressed not only by NSCs and astrocytes, but also by neural progenitor cells.

The weak expression of GFAP and the rareness of GFAP+ fibers spanning the GCL at P0 reflects that aNSCs with typical radial glia-like morphology have not yet established along the hilar-GCL boundary.

2. i) Fig. 2D and Fig. 3D quantify "NSCs in SGZ". How do you define the SGZ at time points before P14 (Fig. 2D), especially P0 and P7?; ii) Fig. 2D suggests that virtually no NSCs (D2+ or D2-) are in the defined "SGZ" at P0, despite the fact that most NSCs at P0 are D2+ (Fig. 2C). Then are the P0 NSCs all outside of the defined "SGZ"?; iii) The definition of "SGZ" at P0 and P7 could be critically important for the conclusion that "D2+ NSCs are a transient population that appears shortly after birth and increases significantly until P10, before declining and returning to nearly P0 levels at P28 (Figure 2D)." It might be more comprehensive to count NSCs in the entire DG region.

Response: We understand the reviewer's concerns and agree that the definition of the SGZ is crucial to correctly interpret our findings. Indeed, a clear demarcation of the SGZ is difficult in newborns. Therefore, we have applied what we consider to be the most objective criteria to distinguish the SGZ from the GCL and hilus in animals below P14. At P0, where the SGZ is not yet established and dNSCs with a process directed towards the molecular layer are distributed across the prospective SGZ and GCL, quantifications were done throughout this entire region that can be distinguished from the hilus due to its higher cell density. From P7 onwards, the SGZ is more clearly formed, defined as tightly packed layer of DAPI+ nuclei that is densely populated with GFP+ cell bodies, indicating the accumulation of NSCs and intermediate progenitors. These criteria are now described in detail in the Methods section of the revised manuscript (page 27, lines 554 ff.), and illustrated in Reviewer Figure 1. To avoid misinterpretation, we have also added a note on where we quantified NSCs at different ages before explaining the results (page 8, lines 158 ff.).

Reviewer Figure 1. Confocal images illustrating the demarcation of the SGZ from P7 to adult, and of the SGZ/GCL in P0. Blue = DAPI, green = nestin-GFP.

Although the pool of NSCs with radial glia-like features is not yet established along the border between hilus and GCL at P0, the few that already settled in the area of the nascent SGZ/GCL express D2. We see no contradiction in this. As the reviewer correctly assumes, most DG-NSCs, including the precursors of aNSCs, reside in the earlier germinal matrices at P0, e.g. the hilar and subpial area, or still migrate along the DMS. While it would be interesting and important to trace the origins of aNSCs further back, this is technically difficult to implement and outside the scope of this study. Quantifications throughout the entire DG would encompass different germinal matrices and thus dilute the findings on the establishment of aNSCs in the SGZ. Moreover, the poorly known cellular heterogeneity in areas outside the SGZ/GCL can lead to misinterpretations of results. We consider these reasons strong enough to focus exclusively on the SGZ (SGZ/GCL in P0), where NSCs with an adult-like phenotype are established early after birth. Importantly, the same criteria were defined and applied for WT and KO mice.

3. The manuscript claims that most NSCs are D2+ before P14 (Fig. 2C), but that a D2+ NSC population is only transiently present in the SGZ during early postnatal development (Fig. 2D). It is possible that at P0 and P7 there are few D2+ NSC in the SGZ because the SGZ has not fully formed. Then it is possible that the decline of D2+ NSCs from P10 to P28 is simply the normal age-dependent decline in actively dividing NSCs (Fig. 3B).

Response: The reviewer is absolutely right, these are indeed the most likely explanations for the transient appearance of D2+ NSCs in the SGZ. The formation of the SGZ is directly related to the accumulation of aNSCs along the border between the hilus and the GCL. At P0, this area contains almost no D2+ NSCs because NSCs with

RGL-like properties are not yet established (Fig. 3D; Nicola *et al.*, 2015; Berg *et al.*, 2019); however, the majority of the few NSCs that can be found express D2 (Fig. 2C). D2 remains expressed in a large proportion of SGZ-NSCs while the pool is rapidly expanding until the end of the second postnatal week (Fig. 3D), explaining the increasing numbers of D2⁺ NSCs until P10 (Fig. 2D). Thereafter, the aNSC pool is established and the fraction (Fig. 2C) as well as the number (Fig. 2D) of NSCs expressing D2 decline sharply. This decline likely reflects both, the cessation of aNSC pool development and the transition of NSCs to quiescence. This is corroborated by the increase in D2⁻ NSCs that continues beyond P10 (Fig. 2D) and the decreasing numbers of Ki67⁺ NSCs (Fig. 3C), whose temporal dynamics mirrors that of D2⁺ NSCs (Fig. 2D).

To avoid misunderstandings due to our wording "...D2⁺ NSCs are a transient population..." which could suggest that we refer to a separate entity of NSCs distinct from other NSCs, we have rephrased the corresponding text passage in the results section (page 9, lines 170 f.).

4. Is the number of Ki67⁺ NSCs at P0 different between WT and KO mice? Do you have the data for the P0 time point in Fig. 3C? It would be best if Fig. 3C could mirror Fig. 3B.

Response: We appreciate the important experimental value of this comment. We have repeated the entire experiment, which now includes P0 data for both, proportions and absolute numbers of Ki67⁺ NSCs. This confirmed that the D2KO has no effect on the proportion of dividing NSCs at either time point. Accordingly, the D2KO effect on the number of proliferating NSCs is similar to that on the total size of the NSC pool, i.e. significant differences only appear from P10 onwards.

5. "To examine whether the lack of aNSCs in the mutant DG was indeed caused by a deficit in their generation instead of premature differentiation or death of dNSCs, we quantified the numbers of astrocytes, neuroblasts and apoptotic cells in the different layers of the developing DG (Figure S3)." This statement is misleading because the density was quantified, not the total numbers. The total numbers of these cell types are much lower in KO because the SGZ surface area is decreased (Fig. 3F).

Response: We regret our lack of clarity in this point. We have exchanged the word 'numbers' by 'densities' on pages 11 and 12. Indeed, quantifying total numbers in this case could be misleading since the DG of D2KO mice is smaller than that of WT mice. The densities provide more appropriate information of relative differences independent of size, showing that there is no disproportional increase in differentiated cells between WT and KO which could explain the lack of NSCs in the KO.

6. Fig. 2B Nestin-GFP staining is incomparable to Fig. 3 Nestin-GFP staining. Staining in Fig. 2B seems like it would be difficult to quantify. Are there better images for Fig. 2B for at least P0 and maybe also P7?

Response: All figures shown in the manuscript represent maximum intensity projections (MIPs) of Z-stacks. At P0, the perceived differences between Fig. 2B and Fig. 3A are likely due to differences in image size. In addition, they might be attributed to two other factors: 1) capturing on different confocal microscopes (Fig. 2B: Zeiss LSM710, Fig. 3A: Leica SP8), and 2) slight differences in the height of Z-stacks used to generate the MIPs, which were adjusted according to the combination of markers

stained. We added this information to the legends of Fig. 2B and 3A. To clarify, the quantifications were not performed on MIPs, but on the 3D Z-stacks, taking into account each individual optical plane (now stated explicitly on page 28, lines 547 f. of the revised manuscript). Please find below magnified single optical planes of the Z-stack used to generate the MIP shown in Figure 2B, demonstrating the quality of the Nestin-GFP staining in the DG in newborns (Reviewer Figure 2).

Reviewer Figure 2. Single optical planes of the Z-stack used to generate the MIP shown in Figure 2B. White = DAPI, green = nestin-GFP. Numbers indicate the position along the Z-axis.

7. "These data demonstrate that both D-cyclins are highly expressed by dNSCs during the time of peak proliferation in the developing DG (Altman and Bayer, 1990a; Bond et al., 2020) and are then downregulated making the NSCs transition to a quiescent adult state (Berg et al., 2019)." The data shown suggest an association between D-cyclin expression in NSCs and the timing of the NSC transition to a quiescent adult state, but they do not provide evidence that the change in D-cyclin expression causes ("makes") the NSCs transition.

Response: We fully agree with the reviewer and corrected the phrasing of this sentence (page 10, lines 185 f.).

8. "Strikingly, mCherry+ aNSCs were only found when RV-SFFVmCherry was injected into the DG but not into the DMS (Figure 4D-E). This result indicates that aNSCs are generated exclusively from precursors located in the DG early after birth, while precursors dividing in the postnatal DMS do not contribute to the formation of aNSCs and adult neurogenesis." To say that adult NSCs are generated exclusively from precursors in the DG early after birth is a bit of an overstatement of the data. The data (Fig. 4) certainly provide evidence that more aNSCs were generated from DG precursors than DMS precursors, but there are still a small number of aNSCs generated from P2 DMS precursors.

Response: The reviewer is correct. We have revised the text and toned down the conclusion to clarify that a few of the dNSCs migrating along the DMS at P2 contribute to the formation of aNSCs (pages 13, lines 259 ff. and page 16, lines 329 ff.).

9. Fate mapping could provide more robust evidence that D2 is required for the generation of adult NSCs. The SFFV-RV-mCherry virus could be injected into the dentate gyrus of wildtype and D2 KO mice at P2 just as in Fig. 4, similar quantifications to Fig. 4D-E could be performed. The hypothesis would be that fewer NSCs would be generated in D2 KO mice.

Response: Unfortunately, we cannot repeat these methodologically demanding experiments in D2KO mice in a reasonable time frame. The double-transgenic mice are no longer in stock and would need to be transferred to Bilbao, where we have the expertise with RV injections in perinatal mice, which would also entail lengthy application procedures to animal welfare authorities. Although we agree that the proposed experiment would strengthen our current findings, we believe that the results shown in Figure 3 provide sufficient evidence that the D2KO impairs the formation of aNSCs. However, we are in the process of creating a conditional mouse line to delete D2 spatiotemporally controlled during DG development and in dNSCs—but that is clearly beyond the scope of this work.

Dear Anja,

Thank you for submitting your revised manuscript to The EMBO Journal. Your study has now been seen by the original referee #3. As you can see from the comments below, the referee appreciates the introduced revisions, but also has a remaining point regarding the quantification method and that it based upon SGZ, but not hilar regions. The referee suggests using an additional marker. Would be good to discuss this issue further.

- When you submit your revised version will you also address the following editorial points:

- You have 6 keywords but can only have 5

- The Data availability section is missing. This is the section to enter accession numbers for generated data in the study. If no data is generated that needs to be deposited in a database then please state: Data Availability: This study includes no data deposited in external repositories.

- COI needs renaming to "DISCLOSURE AND COMPETING INTERESTS STATEMENT"

- Please remove the Authors Contributions from the manuscript. The 'Author Contributions' section is replaced by the CRediT contributor roles taxonomy to specify the contributions of each author in the journal submission system. Please use the free text box in the 'author information' section of the manuscript submission system to provide more detailed descriptions (e.g., 'X provided intracellular Ca⁺⁺ measurements in fig Y')

- Please make sure that the Author Checklist is complete.

- The synopsis image needs to be provided as a JPEG file

- Could you provide source data for Figure EV1F. The CCND3 panel is empty of image noise would be good to double check this.

- We are missing a legend for figure 1D'.

- In the figure legends please indicate what ' $\Omega\Omega\Omega$, $\alpha\alpha\alpha$ ' represents; if this represents p values, please indicate the statistical test used and define these annotations in the legends of figures 1e-f.

- Information related to n is missing in the legend of figures 1e-g; 2f."

- Please indicate what white arrows represent in figure 4a-b"

Looking forward to hearing back from you regarding the remaining issue brought up by referee #3.

With best wishes

Karin

Karin Dumstrei, PhD
Senior Editor
The EMBO Journal

Guide For Authors: <https://www.embopress.org/page/journal/14602075/authorguide>

Use the link below to submit your revision:

Referee #3:

The author have made revision to their manuscript with some new experiments, however, one major question remains to be

addressed.

Quantification methods: The authors based on their quantification on SGZ, but not hilar regions. As pointed out in the previous review, there is no definitive SGZ at P0 and there is large number of GFP+GFAP+ cell in the hilar area. If the authors would like to maintain such quantification, they need to provide evidence that hilar GFP+/GFAP+ cells do not contribute to the aNSC pool. It is an open question whether dNSCs originally located at future SGZ location become aNSCs or dNSCs migrate to SGZ during the formation of the adult neurogenic niche.

GFAP as a marker for NSCs: GFAP clearly is not a good marker for NSCs as they are not expressed in the early dNSCs and it is expressed by astrocytes. Confirmation with another marker, even not a perfect marker, will increase the confidence of the conclusion.

As hinted by other reviewers, I also find the distinction of "dNSCs that simply change their behavior after colonizing the SGZ" and "aNSCs are formed as a distinct population" not very clear. Of course, there many differences between dNSC and aNSCs, D2 dependence being one of them. But there are also many differences for aNSCs in the young adult and during aging, including gene expression, activation rate and division mode (as well as morphology of neurons derived from these aNSCs). Do we call them distinct NSC populations? I suggest the authors just highlight the difference of aNSCs and dNSCs, but avoid the calling them distinct populations (line 342-355).

Dear editors,

Thank you for providing the opportunity to revise and resubmit our manuscript. We addressed all editorial points as well as the comments of referee 3 (see below). In response to the reviewer's comments, we have added a paragraph to our manuscript in which we discuss the limitations and potential caveats of our study. Changes to the text have now been color-coded in orange.

I look forward to your decision and remain with best regards

Anja

Editorial points

Dear Anja,

Thank you for submitting your revised manuscript to The EMBO Journal. Your study has now been seen by the original referee #3. As you can see from the comments below, the referee appreciates the introduced revisions, but also has a remaining point regarding the quantification method and that it based upon SGZ, but not hilar regions. The referee suggests using an additional marker. Would be good to discuss this issue further.

When you submit your revised version will you also address the following editorial points:

You have 6 keywords but can only have 5

Response: We have changed the number of keywords to 5.

The Data availability section is missing. This is the section to enter accession numbers for generated data in the study. If no data is generated that needs to be deposited in a database then please state: Data Availability: This study includes no data deposited in external repositories.

Response: We have added the section accordingly.

COI needs renaming to "DISCLOSURE AND COMPETING INTERESTS STATEMENT"

Response: We have changed the title of this section.

Please remove the Authors Contributions from the manuscript. The 'Author Contributions' section is replaced by the CRediT contributor roles taxonomy to specify the contributions of each author in the journal submission system. Please use the free text box in the 'author information' section of the manuscript submission system to provide more detailed descriptions (e.g., 'X provided intracellular Ca⁺⁺ measurements in fig Y')

Response: We have removed the section Author Contributions.

Please make sure that the Author Checklist is complete.

Response: We have added the information missing from the Author Checklist form.

The synopsis image needs to be provided as a JPEG file

Response: The synopsis is now available as JPEG file.

Could you provide source data for Figure EV1F. The CCND3 panel is empty of image noise would be good to double check this.

Response: Figures 1D to F' were obtained from the Allen Brain Atlas. We now provide the data for the complete, uncropped sagittal brain sections extracted from Allen Brain Atlas as source data for all three D-cyclins.

We are missing a legend for figure 1D'.

Response: We regret the absence of the legend to Figure 1D', which has now been included in the revised manuscript.

In the figure legends please indicate what ' $\Omega\Omega$, $\alpha\alpha$ ' represents; if this represents p values, please indicate the statistical test used and define these annotations in the legends of figures 1e-f.

Response: The information has been added to the legend to Figure 1 (Data information).

Information related to n is missing in the legend of figures 1e-g; 2f."

Response: We regret the absence of this information, which has now been added to the revised figure legend.

Please indicate what white arrows represent in figure 4a-b"

Response: White arrows indicate mCherry+ NSCs. We added this information to the figure legend.

Referee #3:

The author have made revision to their manuscript with some new experiments, however, one major question remains to be addressed.

Quantification methods: The authors based on their quantification on SGZ, but not hilar regions. As pointed out in the previous review, there is no definitive SGZ at P0 and there is large number of GFP+GFAP+ cell in the hilar area. If the authors would like to maintain such quantification, they need to provide evidence that hilar GFP+/GFAP+ cells do not contribute to the aNSC pool. It is an open question whether dNSCs originally located at future SGZ location become aNSCs or dNSCs migrate to SGZ during the formation of the adult neurogenic niche.

Response: We acknowledge the reviewer's observation that the SGZ is not as clearly defined at P0 as in later stages and that many GFP⁺/GFAP⁺ cells are located in the hilus. Indeed, there is evidence that hilar progenitors could contribute to the aNSC pool, as suggested by previous reports (e.g. Namba et al., 2005) and by our RV injections. Although we are aware that studying developmental germinal niches would advance our understanding of aNSC development and the importance of D2, we focused our study strictly on the SGZ for the following reasons:

1. The SGZ is the endpoint of DG development and the only area where NSCs with adult-like features become established (this study; Namba et al., 2005; Nicola et al., 2015). As many others and our own results show, these cells are not present in the hilar region of either the WT or D2KO mice.
2. Current histological techniques do not allow to unequivocally distinguish the precursors of aNSCs from those of glia and neurons in the perinatal hilus. Thus, although the SGZ is not properly defined

at P0, we consider it more appropriate to strictly focus on quantifying NSCs with adult-like properties in the area of the prospective SGZ in order to assess the establishment of the aNSC population in the SGZ over time.

3. We respectfully disagree with the notion that we “need to provide evidence that hilar GFP+/GFAP+ cells do not contribute to the aNSC pool.” As the reviewer points out, it remains “an open question whether dNSCs originally located at future SGZ location become aNSCs or dNSCs migrate to SGZ during the formation of the adult neurogenic niche”, which the current study as many others before cannot clarify. Importantly, the possible contribution of cells from the hilar region does not invalidate the results and conclusions of the present manuscript.

On the other hand, it is also important to mention that the quantification of hilar GFP+GFAP+ cells would not clarify when exactly does D2 become relevant for aNSC formation. In fact, there is a possibility that the basis for the deficit we observe in postnatal KO mice is laid at an earlier developmental stage, e.g., that the division of embryonic aNSC precursors already depends on D2. Potentially, this could be as early as in the VZ or any later germinal zone. Figuring this out is technically challenging. As we propose in our discussion, we need new mouse models that allow time-specific deletion of D2 as well as fate mapping of D2⁺ NSCs, which are a work in progress. Nevertheless, our results strongly suggest that DG development is normal until birth. The possibility that earlier developmental deficits, affecting only hilar precursors, that are manifested later in the perinatal period, and that are enough to justify the lack of aNSCs at later stages, is not supported by our or former data.

In any case, this possibility would not contradict our conclusions that the last divisions forming aNSCs occur within the postnatal DG and that aNSCs are formed in a D2-dependent manner.

GFAP as a marker for NSCs: GFAP clearly is not a good marker for NSCs as they are not expressed in the early dNSCs and it is expressed by astrocytes. Confirmation with another marker, even not a perfect marker, will increase the confidence of the conclusion.

Indeed, the lack of exclusive markers for hippocampal NSCs is a problem that complicates their analysis, and even more so during development. However, we must reiterate that our study is exclusively focused on the NSCs with adult characteristics, for whose identification nestin/nesGFP and GFAP co-expression in tandem with distinct morphological criteria (a radial process extending towards the molecular layer) is the most adequate marker combination available and used by many other studies.

As we noted in our previous response to the reviewer, other markers are even worse at reliably identifying NSCs across different ages. These include Sox2, which labels NSCs, progenitors and astrocytes and is therefore more unspecific than GFAP. Another useful aNSC marker is lysophosphatidic acid receptor 1 (LPA₁), which has almost complete overlap with nestin (NSCs, early progenitor cells) in the adult DG (Walker et al., 2016). However, in the early postnatal DG, LPA₁ is mainly expressed in the hilar region and frequently in neurons and only later in NSCs (Walker et al. 2016; Valcárcel-Martín et al. 2020). S100 β , in combination with nestin-GFP and GFAP, can be used as a negative marker to identify NSCs in the juvenile and adult DG because it is expressed only by astrocytes but not by NSCs. However, this is inapplicable during early postnatal days, when many neuro- and gliogenic progenitor cells express S100 β (Namba et al., 2005). Therefore, we respectfully disagree with the reviewer that the use of other, less specific markers would increase the confidence of our study. The combination of Nestin-GFP, GFAP, anatomical location and morphology are up to this point the best set of criteria to identify NSCs in the dentate gyrus in tissue.

As hinted by other reviewers, I also find the distinction of "dNSCs that simply change their behavior after colonizing the SGZ" and "aNSCs are formed as a distinct population" not very clear. Of course, there many differences between dNSC and aNSCs, D2 dependence being one of them. But there are also many differences for aNSCs in the young adult and during aging, including gene expression, activation rate and division mode (as well as morphology of neurons derived from these aNSCs). Do we call them distinct NSC populations? I suggest the authors just highlight the difference of aNSCs and dNSCs, but avoid the calling them distinct populations (line 342-355).

We are aware of the controversy regarding aNSCs and adult neurogenesis being distinct from or a continuation of their developmental counterparts. We already toned down this discussion in response to the other reviewers, but we think it is very important to address this issue and frame our data accordingly. Although our data do not contradict any of the current models, they provide several lines of evidence for aNSCs being a discrete population that is different from (although likely descended from) dNSCs. In contrast to the notion of a progressive accumulation of changes that occur over time, these differences appear within a narrow time window during the first two weeks of life.

Dear Anja,

I am pleased to inform you that your manuscript has been accepted for publication in the EMBO Journal.

Best regards,

Ioannis
